# Theoretical analysis of Polycomb-Trithorax systems predicts that poised chromatin is bistable and not bivalent

Kim Sneppen [1,3] & Leonie Ringrose [2,3]

Polycomb (PcG) and Trithorax (TrxG) group proteins give stable epigenetic memory of silent and active gene expression states, but also allow poised states in pluripotent cells. Here we systematically address the relationship between poised, active and silent chromatin, by integrating 73 publications on PcG/TrxG biochemistry into a mathematical model comprising 144 nucleosome modification states and 8 enzymatic reactions. Our model predicts that poised chromatin is bistable and not bivalent. Bivalent chromatin, containing opposing active and silent modifications, is present as an unstable background population in all system states, and different subtypes co-occur with active and silent chromatin. In contrast, bistability, in which the system switches frequently between stable active and silent states, occurs under a wide range of conditions at the transition between monostable active and silent system states. By proposing that bistability and not bivalency is associated with poised chromatin, this work has implications for understanding the molecular nature of pluripotency.

[1] Center for Models of Life, Niels Bohr Institute, University of Copenhagen, Blegdamsvej 17, 2100 Copenhagen, Denmark. [2] Integrated Research Institute for Life Sciences, Humboldt-Universität zu Berlin, Philippstrasse 13, Haus 22, 10115 Berlin, Germany. [3] These authors contributed equally: Kim Sneppen, Leonie Ringrose. Correspondence and requests for materials should be addressed to K.S. (email: ksneppen@gmail.com) or to L.R. (email: leonie.ringrose@hu-berlin.de)

Bistable systems can adopt two mutually exclusive stable states[1–3]. Bistability is central to many epigenetic gene regulatory systems[4]. Epigenetic bistability can be visualised as variegation in reporter assays[5–7]. Reporters carrying cis- regulatory elements that recruit epigenetic regulators[5,7], or in which the reporter is placed near to heterochromatin[6], show early random switching and later stable maintenance, resulting in lineages of cells in which the reporter is either off or on, and in which each state is mitotically inherited. Several theoretical models address the potential of bistable systems to maintain epigenetic memory via histone modifications that are dynamic and reversible[8–11]. In these models, a given nucleosome can change its state repeatedly over time, but the system as a whole is stably in an active or silent state, and can survive the disruptions of simulated replication, provided sufficient feedbacks between nucleosomes are present[12,13].

The Polycomb (PcG) and Trithorax (TrxG) group proteins are essential epigenetic regulators that can maintain stable epigenetic memory of silent states (via PcG) and active states (via TrxG) of their target genes over many cell generations[14]. Reporter genes carrying Polycomb/Trithorax Response Elements (PRE/TREs) can show variegation, depending on genomic location and DNA sequence of the PRE/TRE, indicating that the system has bistable properties[5,15]. A potential biochemical basis for this bistability is beginning to emerge (reviewed in[16–18]). The PcG and TrxG proteins modify chromatin, and their biochemistry is exquisitely complex. In the last two decades, over 70 publications have documented biochemical properties of PcG and TrxG proteins and complexes (see Tables 1–3 and Methods). At least ten specific chromatin modifications catalysed by PcG and TrxG proteins have been identified, and the enzymes that add and remove them are well characterised. Furthermore, several self- reinforcing and antagonistic interactions exist, suggesting a potential molecular basis for bistability in the PcG/TrxG system[18] (for full list, see Tables 1-3 and Methods). However the immense complexity of the system means that it also has the potential to adopt more than two states.

Indeed, it has been proposed that a third, bivalent, state is essential in pluripotent stem cells. Bivalent chromatin contains histone modifications catalysed by both PcG and TrxG proteins, and is present in mouse and human ESCs and several other vertebrate cell types[19,20]. Bivalent chromatin is thought to represent a poised or undecided form that is resolved to a stably active or silent form by removal of one or other type of modification upon appropriate signals[21–25]. The best - studied bivalent chromatin form is that containing H3K27me3 and H3K4me3 (histone H3 tritmethylated at lysine 27 or lysine 4). Other bivalent forms include H2A K119 ubiquitination (H2Aub), catalysed by the PRC1 complex, in addition to H3K4me3, RNA Polymerase, and/ or H3K27me3[26–28]. Genes residing in bivalent chromatin are typically silent or expressed at low levels[21–24,28–30]. These genes remain silent upon loss of H3K4me3, and become activated upon removal of H3K27me3 or H2AUb[21,22,25,26,28,31].

The proposition that bivalent chromatin may function as poised chromatin for the PcG/TrxG system is based on indirect evidence. The first studies to identify bivalent chromatin genome- wide did so using separate single – antibody ChIP experiments for each of the modifications in question[21–25]. This approach does not distinguish between true bivalent chromatin and a mixture of different states in different cells. The idea that bivalent marks are essential to pluripotency and resolve upon differentiation was based upon these single-antibody experiments[21,22]. Later studies have used co- ChIP, re-ChIP, mass spectrometry and imaging approaches to confirm that bivalent chromatin containing H3K27me3 and H3K4me3 on the same or adjacent nucleosomes on the same allele does

indeed exist in several cell types[29,30,32–34]. These studies have also given more accurate estimations of the true number of bivalent loci, and how their distribution changes in different cell types[29,30,34]. However, the central question of whether bivalent chromatin is important for defining the poised state and whether it is required for pluripotency is extremely difficult to address experimentally[19,20]. By definition, bivalent chromatin comprises chromatin modifications that are each involved in activation and silencing of many hundreds of genes. Thus any experiment that perturbs bivalent chromatin has a large impact on the PcG/TrxG system as a whole, rendering results difficult to interpret.

To understand the relevance of bivalent chromatin and how it relates to other properties of the PcG/TrxG system, theoretical approaches have immense potential. However, previous theoretical models for PcG/TrxG regulation have not considered bivalent states and the full biochemical complexity of the system[8–11,35]. The greatest obstacle to the integration of individual experimental observations into a coherent whole has been the lack of a comprehensive theoretical framework. In this study, we curated current literature on PcG/TrxG biochemistry, comprising 73 publications, and formalised the resulting information as a dynamic stochastic mathematical model with 144 nucleosome states. Surprisingly, despite the possibility to adopt 144 states, the model has strongly bistable properties, preferring to occupy only the most extreme active or silent states. The model predicts that bivalent states also exist, but that they are present as a dynamic, unstable background population in all system states. Distinct forms of bivalent chromatin preferentially co- occur with active and silent system states. Importantly, midway in the transition between active and silent states, poised chromatin is not bivalent in the model, but is robustly bistable, and differs from monostable modes only in its higher frequency of switching. Furthermore, we show that several published observations strongly support the model predictions. We propose a central role for bistability in PcG/TrxG function, not only for ensuring epigenetic memory but also as a central feature of poised chromatin. Thus, this work has profound implications for understanding the molecular nature of pluripotency and the stability and reversibility of epigenetic states.

## Results

**A comprehensive model of PcG/TrxG regulation.** In the current work, we aimed to investigate emergent properties of the PcG/TrxG system by taking account of its full complexity. To achieve this, we curated all available current literature comprising 73 publications (see Tables 1–3 and "Methods" section), and formalised the observations in a dynamic stochastic model (Fig. 1). The model contains all biochemical properties of the system thus far reported, including the possibility for bivalent states. We nevertheless introduced various simplifications as follows (the model is explained in detail in Methods):

The model contains a reduced number of histone modifications. We use a dynamic stochastic model, formulated in terms of known PcG and TrxG enzymes and nucleosome modifications. We consider four nucleosome modifications: PcG- catalysed H3K27 methylation and H2AK119 ubiquitination, and TrxG-catalysed H3K27 acetylation and H3K4/K36 methylation[14]. The rationale for fusing H3K4 and HK36 methylation, which are catalysed by different TrxG proteins, is based on observations that these proteins interact physically and functionally, frequently colocalise, and that they can recruit each other ([36–39], see Tables 1–3 and "Methods" section for more detail). The model consists of an array of nucleosomes, each of which can carry combinations of these four modifications (Fig. 1a).

**Table 1 Enzymes and histone modifications represented in the model**

| Activity | Enzyme or complex (*Drosophila*) | Enzyme or complex (vertebrate) | Model name |
|---|---|---|---|
| H3K27me3 addition | PRC2 (subunit E(Z)) (H3K27me3)[79,80] | PRC2 (subunit EZH2) (H3K27me3)[77,78] | PRC2 |
| H3K27me3 removal | dUTX[85] | UTX (KDM6A) and JMJD3 (KDM6B)[46,82-84] | UTX |
| H2Aub addition | PRC1 subunit dRING (monoubiquitinates H2AK118)[88] | PRC1 subunits RING1A and RING1B (monoubiquitinate H2AK119)[88-90] | PRC1 |
| H2Aub removal | PR-DUB subunit BAP-1[93] | PR-DUB subunit BAP-1[94] | PR-DUB |
| H3K27ac addition | CBP (acetylates several residues including H3K27)[96] | CBP and p300 (acetylate several residues including H3K27)[40] | CBP |
| H3K27ac removal | RPD3 (deacetylates several residues including H3K27)[96] | NuRD (deacetylates several residues including H3K27)[97] | NURD |
| H3K4me addition | TRX (H3K4me1)[42] TRX (H3K4me2)[105] SET1 (H3K4me3)[103,104] | MLL1 (H3K4me1)[42] MLL1 (H3K4me2)[105] SET1 (H3K4me3)[101,102] | TRXG |
| H3K4me removal | Lid (KDM5 homolog)[112,113]. dKDM2[115] | JARID1A (synonyms: RBP2, KDM5A)[111]. KDM2B[86] | KDM |
| H3K36me addition | ASH1 (H3K36me2)[43,107,108] | ASH1L (H3K36me2)[109,110] | TRXG |
| H3K36me removal | Fbxl10 (synonyms: dKDM2, JHDM1A)[91] | Fbxl10 (synonyms: KDM2B, JHDM1A)[114] | KDM |

**Table 2 Self- reinforcing interactions**

| Molecule | Activity | *Drosophila* | Vertebrate |
|---|---|---|---|
| H3K27me3 | Binds PRC1 | Pc chromodomain binds H3K27me3[116-118] | CBX2 and CBX7 chromodomains (PC homologs) bind H3K27me3[119] |
| H3K27me3 | Binds and stimulates PRC2 | Genetic evidence for similar mechanism in *Drosophila*[121]. | H3K27me3 binds PRC2[120,121]. H3K27me3 stimulates PRC2 activity[121]. |
| H2Aub | Binds and stimulates PRC2 | Binds PRC2[122]. | Binds PRC2[122-125]. Binding stimulates activity[122] |
| H3K4me | Stimulates CBP | H3K4me1 stimulates CBP in H3K27 acteylation[42,96] | |
| H3K27ac | Enhances TRXG binding to chromatin | FSH1 (homolog of BRD4) binds and colocalises with acetylated histones and interacts with ASH1[127,128] | TrxG protein BRD4 binds acetylated histones in vitro and in vivo[142] |
| ASH1 and TRX | Interact physically and functionally | In vitro binding and in vivo colocalisation, genetic interaction[37]. TRX chromatin association depends on ASH1[36] | In vivo colocalisation on single genes[38]. Colocalisation of MLL and ASH1L-catalysed H3K36me2 genome wide. MLL chromatin binding depends on H3K36me2 binding[39] |
| TRXG and CBP | Interact physically and functionally | ASH1 and CBP interact physically and functionally[138]. The TrxG protein BRM is associated with CBP and stimulates the activity of CBP in acetylating H3K27[139] | |
| KDM and NURD | Interact physically and functionally | H3K4 demethylase Lid interacts physically and functionally with RPD3[136]. | H3K4/K6 demethylase LSD1 is a component of the NuRD complex at active enhancers[137]. |

**Table 3 Antagonistic interactions**

| Molecule | Activity | *Drosophila* | Vertebrate |
|---|---|---|---|
| H3K27me3 | Inhibits TRXG binding | | Human SET1 and MLL1 complexes bind poorly to H3K27me3 histones. Catalytic activity is not prevented[134] |
| H2Aub | Inhibits TRXG activity | | Histone H2A ubiquitination inhibits the enzymatic activity of H3 lysine 36 methyltransferases[45]. |
| H3K4/K36me | Inhibits PRC2 activity | H3K4 and H3K36 methylation inhibit PRC2 H3K27 methylation activity[43,44] | H3K4 and H3K36 methylation inhibit PRC2 H3K27 methylation activity[32,44]. |
| PRC1 and CBP | PC inhibits CBP activity | Polycomb (PC, subunit of PRC1) inhibits histone acetylation mediated by CBP by binding directly to the CBP catalytic domain[133] | |
| PRC2 and KDM | Interact physically and functionally | | PRC2 recruits RBP2 (H3K4 demethylase)[111] and LSD1 (H3K4 demethylase)[135] |
| PRC1 and KDM | Interact physically and functionally | Non-canonical PRC1 complex dRAF contains dRING (ubiquitin ligase) and dKDM2 (H3K4 and H3K36 demethylase)[115,91] | Non-canonical PRC1 complex PRC1.1 contains dRING (ubiquitin ligase) and KDM2B (H3K4 and H3K36 demethylase)[86,114,92] |
| TRXG and UTX | Interact physically and functionally | UTX (H3K27 demethylase) is associated with CBP and the TrxG protein BRM[139] | UTX (H3K27 demethylase) is associated with MLL 2/3 (vertebrate homologs of TRX)[46] |

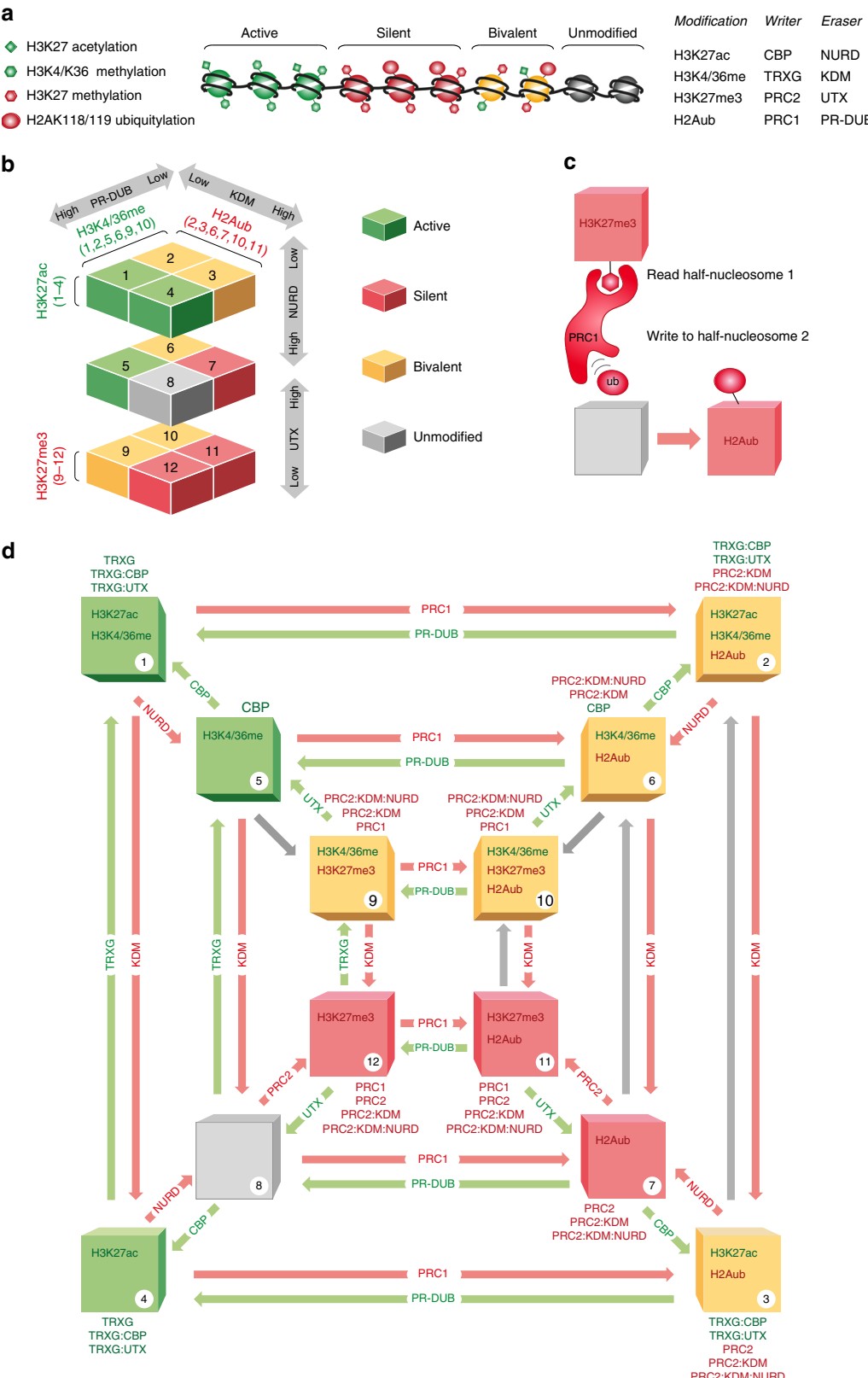

The model is based on half-nucleosomes, and combines them to give whole nucleosomes. The model is formulated in terms of the smallest unit possible, namely the half-nucleosome containing a single copy of H3 and H2A. Whole nucleosomes each comprise two half-nucleosomes, which are paired for the duration of the simulation except during replication. Bivalent nucleosomes are

scored as all whole nucleosomes containing a mix of opposing modifications (Fig. 2).

The model allows combinatorial modifications. Each half-nucleosome in the model can be unmodified or can carry any combination of three modifications, but not all four simultaneously, because H3K27 methylation and acetylation are

**Fig. 1** A comprehensive model of Polycomb/Trithorax regulation. **a** Left: The model considers four histone modifications catalysed by TrxG and PcG proteins. Middle: The model consists of an array of half-nucleosomes, each of which can carry one or more of these modifications, and are combined to give whole nucleosomes (see Fig. 2b). Right: the writers and erasers for each of these modifications are included in the model (see also Table 1). A simplified nomenclature is used throughout as indicated. TRXG indicates both H3K4 and H3K36 methyltransferases. KDM indicates both H3K4 and H3K36 demethylases. **b** There are 12 different possible modification states for each half-nucleosome (containing one copy of H3 and one of H2A), shown in a three-dimensional geometry. Adjacent states in horizontal and vertical directions are different from one another by one modification. Modifications in common for given half-nucleosome states are indicated. Grey arrows indicate the direction in which the system state is pushed by raising or lowering the activities of the complexes indicated. **c** Model logic. Half-nucleosome 1 is selected at random from the array, and recruits a complex (reader) depending on modifications. The recruited complex attempts to modify (write to) a second randomly selected half-nucleosome. **d** Full model. The diagram shown in (**a**) is viewed from above. Half-nucleosome states are numbered 1-12 with modifications as shown. Writers and erasers: Coloured arrows indicate transitions between half-nucleosome states for half-nucleosome 2. Red: towards silencing, green: towards activation. Complexes responsible for each transition are indicated on the arrows. Grey arrows: if the complex responsible for this transition is inhibited by an existing modification on half-nucleosome 2, the transition occurs only via the direct conversion parameter beta (see Methods). Readers: Complexes that can be recruited by half-nucleosome 1 in a given state are shown above each state, with red or green text indicating complexes that favour silencing or activation respectively. Compound complexes (e.g., TRXG:CBP) are included where evidence exists for physical interaction (see Methods and Tables 1–3 for details)

mutually exclusive on the same H3 tail[40–42]. The model thus allows 12 possible modification states for each half-nucleosome, including the unmodified state (Fig. 1b, d). Each of these 12 half-nucleosomes can be paired with any other, giving 144 possible whole-nucleosome modification states (Fig. 2).

Finally, we do not include transcription. The model deals only with the enzymatic modifications of nucleosomes themselves, and does not include the interactions of these histone modifications with polymerase, or with transcription itself[10,11,26,28]. However, for simplicity, we designate PcG-catalysed modifications (H3K27me and H2Aub) as silent, and those catalysed by TrxG (H3K4/K36me and H3K27ac) as active. We further designate system states (the average modification state of the whole array) as active or silent when dominated by one or other type of modification. This nomenclature does not imply any claim to the transcriptional state of the locus, and whether transcription is a cause or consequence of histone modification.

Each of the modifications described in the model is reversible, and the complexes or enzymes that catalyse their addition (writers) and removal (erasers) have been identified in flies and vertebrates (Fig. 1a, right;[14], see Methods and Tables 1–3 for detailed references). Figure 1b represents the 12 possible half-nucleosome states in a three-dimensional geometry, in which adjacent states differ from one another by a single modification. Figure 1d shows this same geometry viewed from above, showing the writers and erasers for transitions between states. The exquisite complexity of PcG and TrxG chromatin comprises several layers. First, there are numerous examples of nucleosome modifications that stimulate or recruit specific writers and erasers. For example, H3K27me3 can enhance the binding of both PRC1 and PRC2 ([14], see "Methods" section and Tables 1–3 for full list). We formalised these observations in the model by imposing appropriate positive feedbacks, whereby existing nucleosome states in the array affect the probability of new modifications to the array. These feedbacks are implemented at the level of cross-talk between two half-nucleosomes: Half-nucleosome 1 is selected at random from the array (Fig. 1c). Specific modifications on half-nucleosome 1 can recruit readers (text above each state in Fig. 1d). Half-nucleosome 2 is selected at random. A reader recruited by half-nucleosome 1 can add (write) or remove (erase) a single modification on half-nucleosome 2, changing its state by one step (see Fig. 1c for example). Any half-nucleosome in the array can stimulate the conversion of any other, except itself.

Second, some histone modifications have been shown to inhibit the activity of specific complexes (PRC2 is inhibited by H3K4/K36me[32,43,44], TRXG enzymes are inhibited by H2Aub[45], see Tables 1–3). This was implemented in the model so that a given writer cannot modify a half nucleosome carrying a modification

that inhibits it (Fig. 1d, grey arrows; see Methods for details). Third, several physical interactions between proteins with different enzymatic activities have been described, for example TRXG proteins recruit the H3K27me3 demethylase UTX[46]. The model explicitly describes all compound complexes for which we found evidence (e.g., TRXG:UTX Fig. 1d; see Tables 1–3 and "Methods" section). In reality these interactions may be highly regulated[19,20]. In the model their presence or absence can be simulated by adaptation of specific parameters.

Finally, there is ample evidence that recruitment of PcG and TrxG proteins occurs not only via existing histone modifications but can also be achieved by independent means, including direct specific and non- specific DNA binding (reviewed in ref. [17], for non- specific DNA binding see refs. [47–50]). In addition, histone modifications can be erased not only by enzymes but also by any process that removes or remodels nucleosomes themselves[8]. For simplicity we designate all such events in the model as direct conversions, meaning that they are independent of recruitment of an enzyme by existing histone modifications. The rate of direct conversion can be adjusted for each transition separately. However, in the absence of quantitative data, we use a single parameter 'beta' for most direct conversions (see "Methods" section). Remarkably, we found that evidence exists for every enzymatic reaction (Table 1a), and for the majority of self-reinforcing and antagonistic interactions (Table 1b) in the model in both flies and vertebrates. Thus the model is potentially equally relevant to both (reviewed in[17], see "Methods" section for detailed references). In summary, the model unifies current literature on the biochemistry of Polycomb/Trithorax regulation into a single coherent framework.

**Bivalent nucleosomes fall into several categories**. Bivalent nucleosomes have been defined experimentally as those that contain opposite modifications on the same nucleosome, but not necessarily on the same histone[29,32,34]. In order to allow evaluation of full nucleosome modification states in the model, we assigned each half- nucleosome in the array to a partner, so that each pair of nucleosomes represents a full nucleosome with two H2A and two H3 tails (Fig. 2a). Figure 2b shows the 144 full-nucleosome modification states that emerge from all possible pairings of half-nucleosomes. We classify nucleosomes into categories according to the proportion of active and silent modifications they contain, giving 15 forms that each contain only active (dark green, Fig. 2b) or only silent (dark red) modifications, and one that is unmodified (grey, Fig. 2b). The remaining 113 forms contain a mixture of active and silent modifications and are thus bivalent (light green, yellow, and orange, Fig. 2b). Interestingly, these bivalent nucleosomes fall into three categories: those

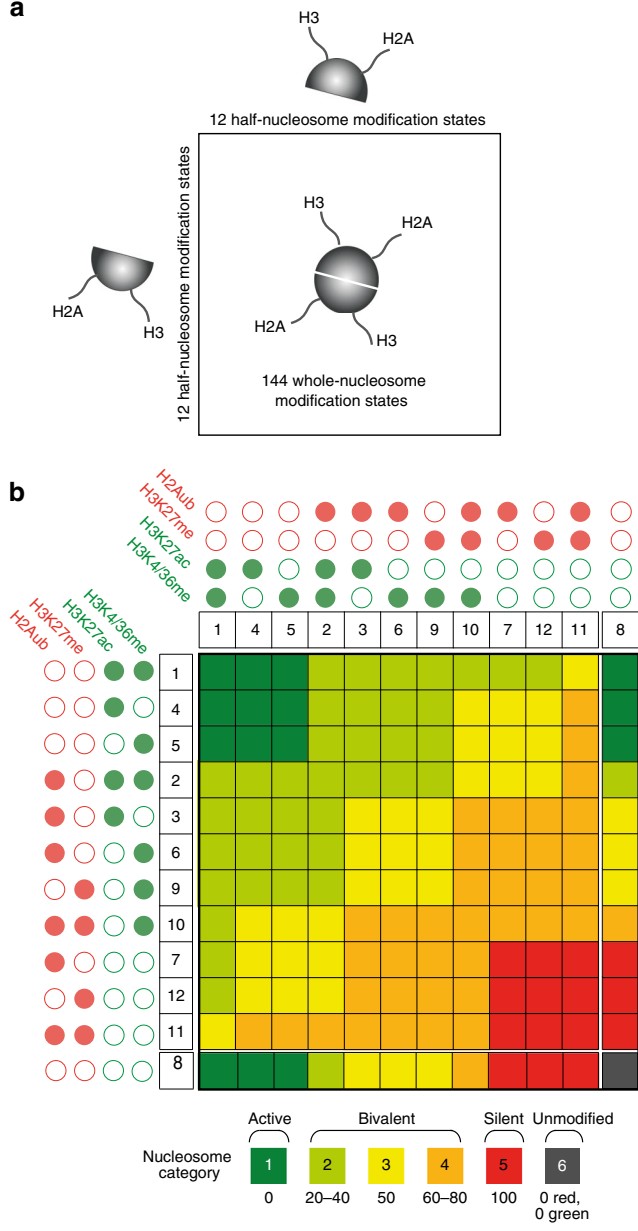

**Fig. 2** 12 half-nucleosomes combine to give 144 whole nucleosomes. **a** Each half-nucleosome contains a single H3 and a single H2A tail, each of which can be modified as shown in Fig. 1d, giving 12 possible different half-nucleosome modification states. Each of these half-nucleosomes can be combined with one of 12 opposite halves, giving 144 possible whole-nucleosome modification states. **b** Half-nucleosomes are shown above and beside the figure with numbers corresponding to Fig. 1b, d. Modifications are indicated as closed circles. Red: modifications associated with silencing; green: modifications associated with activation. Whole nucleosomes are colour coded and assigned to categories according to the proportion of active and silent modifications they contain. For simplicity, this is represented as % red modifications in the colour chart below the plot. Note that when considering whole nucleosomes, we define bivalent nucleosomes as all those in categories 2, 3, and 4, which carry any combination of opposing marks, on the same or different half- nucleosomes

that contain mostly active marks (category 2, light green, Fig. 2b), those that contain mostly silent marks (category 4, orange, Fig. 2b), and those that contain a balanced mix of both active and silent marks (category 3, yellow, Fig. 2b). Specific individual

modifications and pairs of modifications occur preferentially but not exclusively in each of these bivalent categories (Supplementary Fig. 1). For clarity, we henceforth refer to these bivalent categories as active bivalent, silent bivalent, and balanced bivalent, without implying any assumptions as to their effects on transcription.

**Poised chromatin is robustly bistable and minimally bivalent.** To address bivalency and bistability we simulated changes in the composition of the nucleosome array over time (see "Methods" section). Since quantitative information on enzymatic rates is not available for the majority of reactions in our system, we tested the model under a range of different parameter combinations. We first varied the rate of H3K27 deacetylation by NURD in 2-fold steps and plotted a time course for each condition (Fig. 3a–c). We observed that intermediate situations between active (category 1: dark green) and silent (category 5: dark red) cases are bistable in the sense that the system periodically transits between the two extremes (Fig. 3b). This bistable intermediate is not dominated by any of the bivalent categories 2–4 (light green, yellow or orange), but consists of either one or the other extreme modification state for long periods of time. A change of only two-fold in the rate of H3K27 deacetylation in either direction is sufficient to switch the system to a stably active (Fig. 3a) or stably silent (Fig. 3c) state.

To examine the occupation of the 144 possible individual nucleosome modification states (Fig. 2b), we plotted the time-averaged occupation of each nucleosome state on a landscape, showing frequently occupied states on the peaks and rare states in the valleys (Fig. 3d–f). Situations with active or silent state dominance each show a single peak. When active, the system gravitates towards fully modified whole nucleosomes, in which both H3 tails contain both H3K27ac and H3K4/36me (extreme green corner in Fig. 3d). When silent, the system tends towards fully silent nucleosomes, containing H3K37me3 and H2Aub on both half- nucleosomes (extreme red corner in Fig. 3f). This tendency towards fully modified nucleosome states is caused by the strong feedbacks in the system. The intermediate situation with bistability is reflected in an average landscape with two peaks at the extreme corners (Fig. 3e). This analysis demonstrates that a two-fold change in a single enzyme rate is sufficient to flip the system state and affects the distribution of all nucleosome modifications (even those that do not depend directly on that enzyme). We observed similar switching behaviour and category distributions upon two-fold variation of parameters representing each of the eight enzymes in the model (Supplementary Fig. 2a). Thus the system has bistable properties and can be flipped to either extreme state (dominated by fully active or silent nucleosomes) by multiple different small perturbations. We did not observe any condition in the above analysis in which any of the bivalent nucleosome categories 2–4 dominated (Fig. 3g–i, Supplementary Fig. 2b).

To examine system behaviour for a wider range of parameters we systematically varied pairs of parameters and examined the time-averaged system state for each condition (Fig. 3j, k, Supplementary Fig. 3). This analysis showed that for a wide range of parameter combinations, the system recapitulates the features observed above (Fig. 3a–c) with an intermediate between an active and a silenced system that is bistable, and an absence of dominant bivalent states (Fig. 3j, Supplementary Figs. 3a and 4). To challenge the system further, we simulated the effects of replication as described previously[8], by resetting each half-nucleosome to the unmodified state 8 with 50% probability at regular intervals (Fig. 3k, Supplementary Fig. 3b). Interestingly, replication had no effect on the stably active states (compare green zones in Fig. 3j, k, and Supplementary Fig. 3a and b), nor

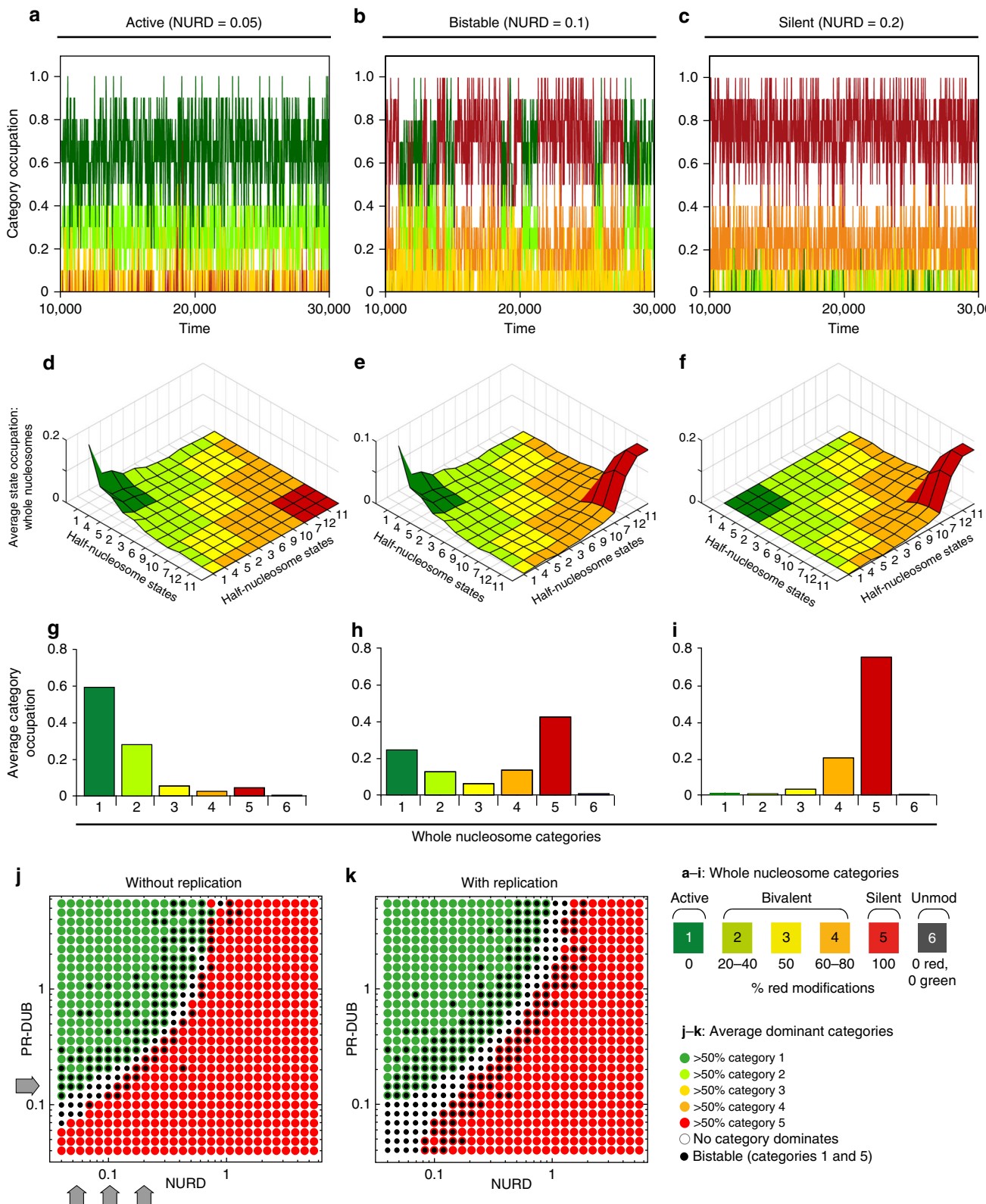

**a** Active (NURD = 0.05)
**b** Bistable (NURD = 0.1)
**c** Silent (NURD = 0.2)

**j** Without replication
**k** With replication

**a–i**: Whole nucleosome categories

| Active | Bivalent | | | Silent | Unmod |
|---|---|---|---|---|---|
| **1** | **2** | **3** | **4** | **5** | **6** |
| 0 | 20–40 | 50 | 60–80 | 100 | 0 red, 0 green |

% red modifications

**j–k**: Average dominant categories

- >50% category 1
- >50% category 2
- >50% category 3
- >50% category 4
- >50% category 5
- No category dominates
- Bistable (categories 1 and 5)

on the most stably silent states (bottom right of red zones in Fig. 3j, k, and Supplementary Fig. 3a and b). Thus in the model, sufficient feedbacks exist for epigenetic memory of both active and silent chromatin states under a wide range of conditions. Surprisingly however, silent states near the transition zone were weakened by replication, with a correspondingly wider parameter regime for bistable transitions between the extreme states (Fig. 2k,

Supplementary Fig. 3b). In general the bistable mode was favoured by replication under these conditions, with several parameter combinations that gave no bistability in the absence of replication, becoming bistable when replication was added (compare white areas at bottom left of plots in Supplementary Fig. 3a and b). At higher replication rates we observed a similar survival of extreme active and silent states, and loss of silent states

**Fig. 3** The model adopts active, silent, or bistable modes. **a–c** Simulated time course of dynamics of a system of $N = 20$ half nucleosomes with 2-fold variation in the rate of removal of H3K27ac by NURD. For all three plots, the rate of all transitions in the model is 1.0, with the exception of NURD as indicated, beta = 0.1, and PR-DUB = 0.15. Time is measured as number of recruitment attempts per half-nucleosome (see "Methods" section for more detail). Y-axis shows proportion of whole nucleosomes in each of categories 1–5, according to the colour scale given on the bottom right of the figure. **d–f** Time-averaged landscapes for the three cases, shown in terms of the probability to find a whole-nucleosome in any of the 144 modification states, averaged over a simulation of 50,000 time units. Half- and whole-nucleosome modification states are arranged and colour coded as in Fig. 2b. The vertical scale gives average proportion of whole array in a given modification state. **e** The bistability shown in (**b**) is reflected in the two peaks at the extreme corners of the landscape. **g–i** The data from (**d**) to (**f**) are summarised in terms of total average occupation of each of categories 1–6. **j** System behaviour upon change in the rate constants NURD and PR-DUB with all other parameters fixed as in (**a–f**). Each simulation was performed for 50,000 time units and average system state over the entire time course was calculated. Red dots: category 5 (silent) dominates (average occupancy of category 5 nucleosomes larger than 50%). Green dots: category 1 (active) dominates. Note that all five modification categories were scored (see legend) but categories 2–4 did not dominate under any condition. Black circles: bistability, defined as multiple transitions between situations with more than 60% category 5 nucleosomes and those with more than 60% category 1 nucleosomes, each of which has an average lifetime of 40 time units. Grey arrows: positions for PR-DUB and NURD values as in (**a–i**). **k** As for (**j**) but with replication at every 20 time units, simulated by resetting each half-nucleosome to the unmodified state with 50% probability

near the transition zone, and a softer transition in which bistability was lost, and the system switched more frequently between short-lived active and silent states (Supplementary Fig. 3c). Importantly, bivalent states (nucleosome categories 2–4) remained rare under all of the conditions tested, and never became stably dominant in the transition zone (Supplementary Figs. 3 and 4). Thus in the model under these conditions, the transition between stable active and silent states typically passes through bistability and not bivalency.

Indeed it was very difficult to find parameters for which the bivalent states became dominant. We found dominant bivalent states when beta, NURD and PR-DUB were all very small (of order 0.005; Supplementary Fig. 5). Examination of categories and specific modifications showed that under these conditions, the system essentially becomes blocked in a state containing H3K27ac and H2Aub, as the rates of removal of these modifications are very low (Supplementary Fig. 4b–g). Similar simulations in which beta was reduced to 0.005 and the rate of removal of H3K4 and H3K27 methylation by low KDM and UTX rates (of order 0.005) resulted in a dynamic bistable system that switches rapidly between silent (category 5) and silent bivalent (category 4) system states with a predominance of bivalent nucleosomes containing H3K4/K36 and H3K27 methylation (Supplementary Fig. 5h–n). We did not find a parameter combination in which category 4 or category 2 stably dominated the system (Supplementary Figs. 4 and 5). We conclude that except under extreme conditions, the model avoids dominant bivalent states, preferring to pass via bistability in the transition between active and silent states. Thus in the model, poised chromatin is robustly bistable and minimally bivalent.

**Active and silent chromatin contain distinct bivalent subpopulations**. Bivalent chromatin containing opposite modifications on the same nucleosome has nevertheless been observed at many PcG target loci in sequential ChIP and co-ChIP experiments[29,30,33,51] and by single molecule imaging[34]. Thus we asked whether our model is consistent with these experimental observations. Although the model predicts that dominant bivalent nucleosome states are difficult to maintain, we did observe a substantial background of bivalent nucleosomes in all simulations, representing up to 40 % of total nucleosomes an any given time point (see light green and orange curves in Fig. 3. a-c). Interestingly we found that the prevalent bivalent nucleosome category changed systematically with system state (Fig. 4). The active system state was accompanied by a background of the active bivalent nucleosome category 2 (Figs. 3g, 4e, f). In contrast, the bivalent background in the silent system mainly comprised

the silent bivalent nucleosome category 4 (Figs. 3i, 4h, i). This preference of active and silent system states for specific active and silent bivalent subtypes held true regardless of which parameter was used to switch the system (Supplementary Fig. 2b). The same subtypes co-occur with active and silent states within the bistable regime and are thus seen as an average in Fig. 3h.

To evaluate whether active and silent system states show a preference for specific bivalent modifications, and to enable comparison with experimental data on pairwise combinations of modifications, we examined four possible pairwise combinations of opposing marks on the same nucleosome (Fig. 4, see also Supplementary Fig. 1). This analysis showed that bivalent H3K27me/H3K27ac nucleosomes are rare, and are depleted from both active and silent system states (Fig. 4c). This is consistent with experimental observations[29,32,40,52]. Furthermore, bivalent nucleosomes containing H2Aub and H3K4/K36me are abundant, are enriched at the border of the active zone, and present in both active and silent system states across a wide range of parameters (Fig. 4d). This is consistent with the observation that this combination of marks can accompany a wide range of gene expression levels[26,28]. In summary in the model, neither of these types occurs exclusively with active or silent system states.

In contrast, across the active parameter regime, the dominant monovalent active nucleosomes (category 1, Fig. 4e), are consistently accompanied by a background of nucleosomes carrying H2Aub and H3K27ac. These are depleted in the silent parameter regime (Fig. 4g). The silent system states (category 5, Fig. 4h) contain a background of methylated H3K27 and H3K4/K36, which are depleted from the active parameter regime (Fig. 4j). Similar distributions of these bivalent nucleosome types with respect to active and silent system states were consistently seen across all parameter combinations tested (Supplementary Fig. 6).

Each bivalent form can also be depleted by extreme parameter conditions that render the system fully active or silent and thus disfavour bivalent forms. For example, bivalent nucleosomes containing H2Aub and H3K27ac are progressively lost from active chromatin as PR-DUB levels increase, because H2Aub is removed (Supplementary Fig. 6b, panels 2 and 5). Likewise, nucleosomes containing H3K4/36 and H3K27me are progressively lost from silent chromatin as KDM levels increase, because H3K4/K36 is removed (Supplementary Fig. 6c, panels 3 to 5). Bivalent chromatin containing H3K27me3 and H3K4me3, with or without additional H2Aub, has been extensively studied experimentally[21–26,28–30]. In contrast, the co-occurence of H3K27ac and H2Aub on the same nucleosome has not been investigated experimentally to our knowledge.

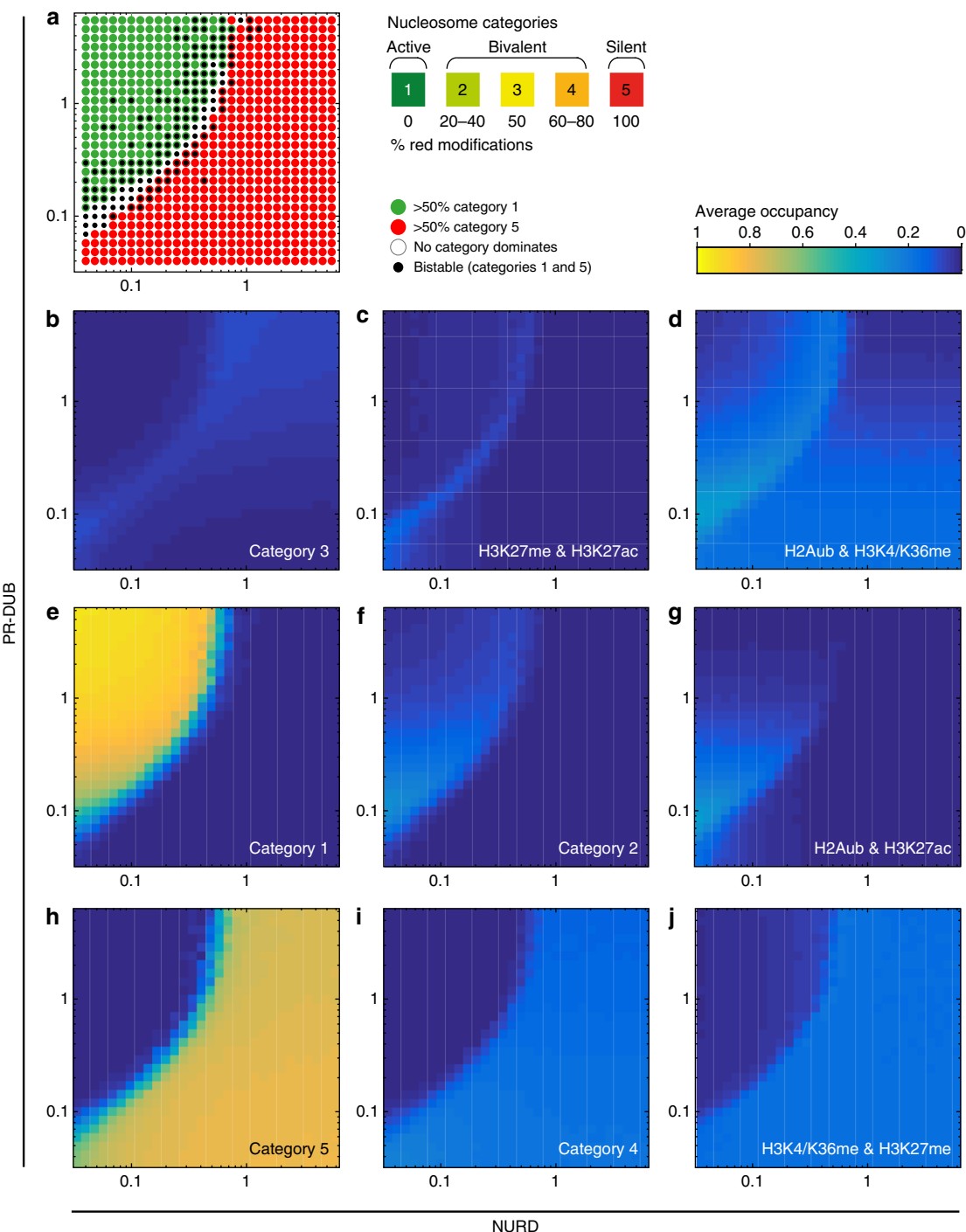

**Fig. 4** Active and silent chromatin contain distinct bivalent chromatin types. **a** Data from Fig. 3j. **b–j** For the same parameter values, average occupancies over the simulated time course for different whole-nucleosome categories as indicated are shown on the same colour scale (right). (**c**, **d**, **g**, **j**) Probability to find bivalent whole nucleosomes carrying both modifications as indicated

In summary this analysis shows that in the model, distinct forms of bivalent chromatin exist as subpopulations within active and silent chromatin as a result of dynamic transitions within the system.

## Discussion

We have modelled the PcG/TrxG system based on currently available knowledge of its biochemical properties. On the basis of this analysis, we propose that bistability is more likely to

represent the poised situation than bivalency (Fig. 5a, b). Stable active and silent system states are nevertheless dynamic, and each contain a background of active or silent bivalent nucleosomes (Fig. 5c).

Our model unites a large body of literature into a coherent unifying framework (Fig. 1), comprising 144 different potential nucleosome modification states. The model is thus more complex than several previous models for PcG/TrxG regulation or epigenetic regulation in general, which typically use the simplest model structure that recapitulates experimental observations and

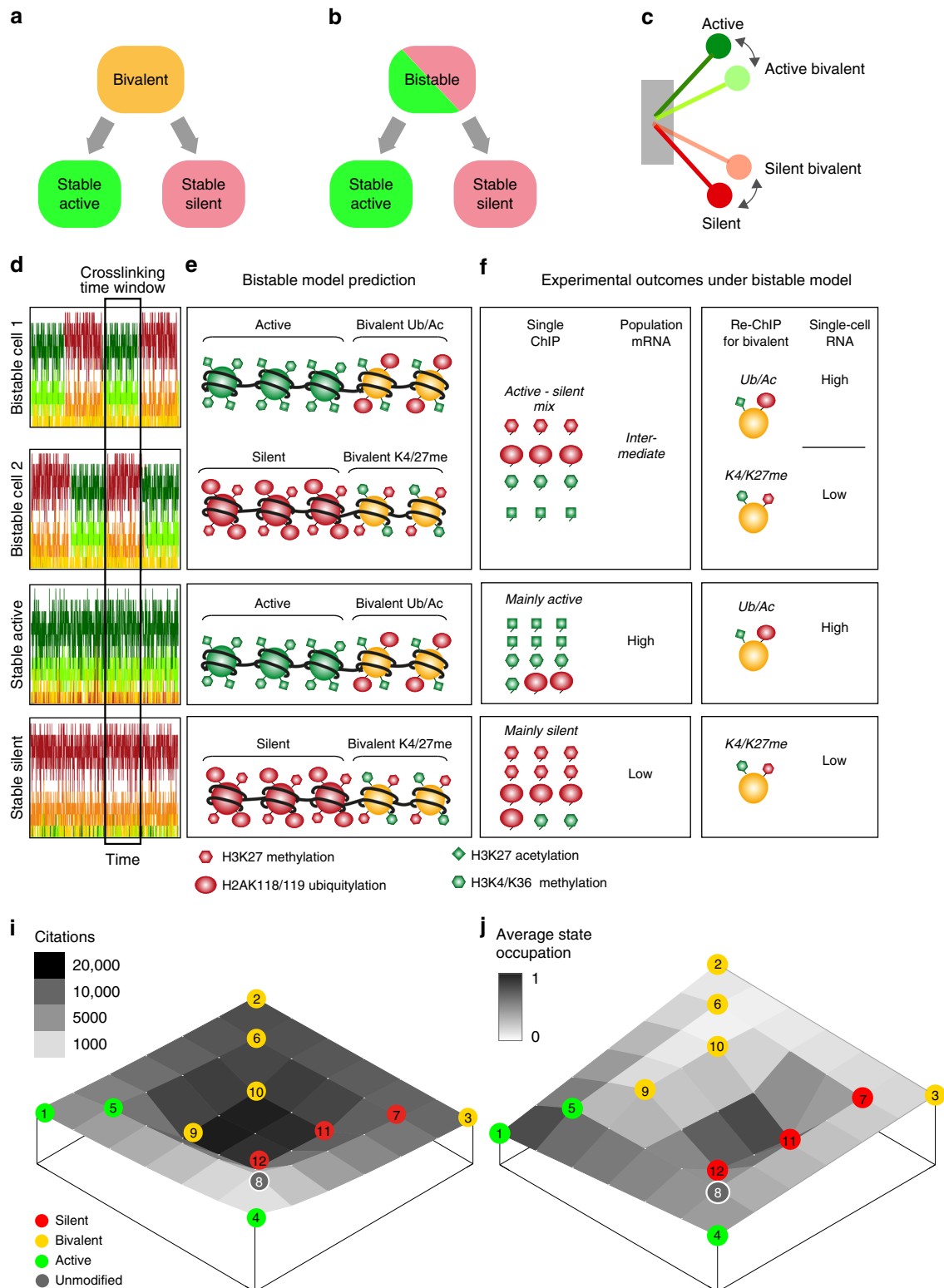

achieves bistability[8–11,35]. In these examples, the system was reduced to 3 states (active, silent and intermediate). This simplifying approach has been designated bottom-down modelling: "Given a model that captures main features of the data, one should persistently strive to simplify it while still capturing the phenomenon."[12]. These simplified models are immensely powerful for understanding unifying principles, but they necessarily make assumptions about the number of states that exist, and the requirement for bistability. Here, we took a bottom-up approach,

to ask: given all the known individual features of the system, and without any assumptions about bistability, what properties emerge, and do they help to understand data that is not captured by the simplified models? PcG/TrxG regulation receives much attention in the literature, with over 600 publications per year since 2013. In the absence of a coherent theoretical framework, it has been difficult to place this large amount of information into the context of the system as a whole. Our model enables this.

**Fig. 5** Reconciling bivalent and bistable chromatin. **a** Previous models propose that bivalent chromatin, containing H3K27me3 and H3K4me3, is poised for activation or silencing in pluripotent cells, and is resolved to an active or silent state upon differentiation. **b** We propose that the poised, pluripotent state is bistable, not bivalent. **c** Dynamic bistability. In the model, in active chromatin, nucleosomes switch dynamically between fully active and active bivalent configurations. In silent chromatin, nucleosomes alternate between fully silent and silent bivalent configurations. **d** Schematic time course plots of bistable model in different modes. Crosslinking window: a ChIP experiment can capture bistable chromatin in different states in different cells and will thus represent a mixture of the active and silent states. **e** Predictions of the bistable model for a single locus. The model predicts that distinct types of bivalent nucleosomes co-occur with active or silent chromatin. **f** Single ChIP and population mRNA analysis would give identical results for both the bivalent and bistable models. **g**, **h** Re-ChIP (i.e. sequential or concurrent ChIP with two antibodies) and single-cell mRNA analysis would give different results under the bivalent (**g**) and bistable (**h**) models. **i** Epigeneticist landscape. Citation frequency for each of the 12 half-nucleosome states shown in Fig. 1d is plotted as – log of total citations of original publications that address a state or transition (see "Methods" section for details and literature). Rarely cited and less abundant publications are shaded light and occur on the hills. Highly cited publications are shaded dark and occur in the valleys. Publications relating to silent or bivalent states that contain H3K27me3 (half-nucleosome states 9–12) are the most abundant and highly cited. **j** For comparison, a similar landscape plotting – log of average occupancy of each of the 12 half- nucleosome states under bistable conditions as in Fig. 3b. Rare states are shaded light and occur on the hills. Frequently occurring states are shaded dark and occur in the valleys. The density of research relating to each state is disproportionate to the predicted occurrence of each in the model system

There are several potential limitations that could complicate the interpretation of the model predictions. First, despite its complexity, the model is still vastly simplified: for example, we do not distinguish between different degrees of lysine methylation (mono- di or tri)[53], we do not consider the interaction of nucleosome states with transcriptional processes[10,11] or the distance between nucleosomes in the array[8,54,55]. These processes could all potentially affect the outcome of simulations. In particular, we treat each half-nucleosome in the array as equally likely to interact with its nearest neighbour as with any other in the array. We reason that since the model array is 2kb in size, this is a realistic approximation. PRC1-bound chromatin arrays have been shown to be highly compacted over similar distances[56] and ChIP-seq peaks of PcG and TrxG protein binding and the modifications they catalyse are typically at least 1-2kb in size[25,57,58]. However, some proteins and modifications do spread over much longer distances of several tens of kilobases[25,59]. Several theoretical and experimental studies have examined the effects of distance between nucleosomes in large modelled arrays, and the phenomena of spreading and looping[8,54,55,60,61]. It will be interesting in future to extend the model to address larger domains, for example by modulating the probability that one nucleosome affects others in the array.

A second potential limitation is the lack of available quantitative information. Although the affinities of several components for their binding partners has been measured (reviewed in refs. [18,62] see also ref. [50]) and for a few components the absolute concentrations have been measured in specific cell types[63,64], the catalytic conversion rates of the enzymes involved are largely unknown. If quantitative data become available, this will allow selection of parameter values. However even in the absence of such data, the model has immense value in defining system properties across a large parameter space, and in predicting the effect of experimentally tractable interventions, for example varying the quantity of any specific enzyme in the system (Supplementary Figs. 2 and 3).

Finally the model is unlikely to be complete: the high rate of research in the PcG/TrxG field means that new modifications, interactions or enzymes will undoubtedly be discovered in the future. Indeed, although no readers of bivalent marks have yet been reported in *Drosophila* or vertebrates, the first bivalent reader (H3K27me3/H3K4me3) has recently been reported in *Arabidopsis*[65]. Unfortunately many other components of the system are not as well characterised in plants as they are in *Drosophila* and vertebrates. However, we reason that within the framework we have established, new components can readily be included as they appear in the literature. Thus, the model is a valuable and evolvable tool for formalising the PcG/TrxG

literature, and can be applied to any organism for which there is sufficient information.

We show that the PcG/TrxG model system is highly bistable. Remarkably, despite the possibility to adopt 144 different states, and despite that fact that we made no assumptions about bistability in the model, one of the most important properties to emerge from the model system is robust bistability. Under a wide range of parameter combinations, the model gravitates toward one or other of the extreme active or silent states (the fully modified active and silent states). There is also a broad parameter regime between these two extremes in which the system is bistable, transitioning rapidly between active and silent states, and can be readily pushed towards one or other extreme stable state by a small change in a single parameter (Fig. 3, Supplementary Fig. 2). Bistability emerges from the model because of multiple cooperative interactions that stabilise each state, and is reinforced by the antagonistic relationships between opposite states (Fig. 1, Tables 1–3).

If the PcG /TrxG system is simply bistable, why is it so complex? The system contains multiple feedbacks that contribute to its robustness. We propose that this complexity contributes both to stability and flexibility. Under conditions that place the system in a stably active or silent state, there is a wide range of values (over several orders of magnitude) for any given parameter pair, that do not destabilise that state, even in the face of rapid replication (Supplementary Fig. 3). Thus, the system has potential for extremely robust memory of both active and silent states that can withstand substantial fluctuations in the activities of its components.

However, the complexity of the system also offers opportunities for flexible regulation. We predict that several different single perturbations can flip the system towards activation or silencing if it is in or near the transition zone (Supplementary Figs. 2 and 3). The activities of different system components may vary globally in different cell types, or locally due to recruitment to specific loci. Each component may be highly regulated[20] and recruitment will also depend on local DNA sequence[14,66]. These observations have important implications for understanding epigenetic memory[14], the consequences of misregulation of PcG/TrxG proteins in disease[67], and the effects and side effects of therapeutic interventions based on inhibition of enzymatic activities[68].

The model predicts that poised chromatin is not bivalent but is robustly bistable (Fig. 5a, b). We used the model to examine the nature of poised chromatin in the transition between active and silent states. Except under extreme conditions (Supplementary Fig. 5) bistability and not bivalency is systematically predicted in the transition regime in which the system switches between active

and silent states (Supplementary Fig. 3). Thus we propose that bistability may be an essential feature of poised chromatin, differing only from the extreme monostable states in its higher frequency of switching. Previous theoretical studies of bistable epigenetic systems have focused on the importance of bistability for long- term epigenetic memory of stable chromatin states in determined cells[8,9]. Our work raises the intriguing possibility that bistable chromatin may in fact be a key molecular feature of poised PcG/TrxG targets in pluripotent cells.

How can this be tested? ChIP experiments cannot detect bistable chromatin, as they examine average enrichments for a population of cells at a given time point (Fig. 5d-f). Instead, analysis of transcriptional noise in single cells may give some insights. Although we are cautious not to equate chromatin state directly with transcriptional state, we do expect some correlation. In this context, the model makes an important testable prediction, namely that a bistable locus would show fluctuations or bursts of transcription over time, to a greater extent than one that is stably active or silent. This would manifest in high cell-to-cell variation in mRNA levels (Fig. 5f). This prediction is consistent with several recent studies, showing that specific classes of mammalian PcG targets do indeed show high cell-to-cell variation in single-cell RNA-seq experiments[69,70], that this variation is greater than that of non- PcG target genes[69] and greater than that of stably active or stably silent PcG target genes[70]. Furthermore, several recent studies based on single-cell imaging of PcG- regulated reporter gene activity have shown that regulation occurs in an all- or none fashion, so that the fraction of cells rather than the amount of gene expression is quantitatively regulated[7,9,35]. Further exploration of the interplay between chromatin states and transcriptional output will be of great interest in future. It will also be important to assess bistability using techniques that capture dynamics in real time[35].

The model further predicts that poised bistable chromatin is influenced by replication. In the model, the transition regime is broadened upon replication, and bistable switching becomes more frequent with replication, with bistability being lost only upon very rapid replication (Supplementary Fig. 3b, c). The inverse relationship between cell cycle length and stability of epigenetic memory has been noted in several theoretical studies[8,10]. Here we propose that the fast-switching bistability induced by replication may in fact be an essential feature of chromatin plasticity in rapidly cycling cells such as pluripotent and cancer cells, by conferring a naïve state on PcG /TrxG target genes.

The model predicts that bivalent chromatin is difficult to maintain as a stable dominant state, except under extreme perturbations (Supplementary Fig. 5). This is consistent with recent theoretical studies, showing that in a system with two opposing modifications, bivalent chromatin emerges only upon perturbation of the system[71], or when parameters that favour bistability are set to 0[72]. However, our model also predicts that without perturbation, and when all states are allowed, bivalent chromatin is present as a relatively abundant sub-population for specific parameter regimes in both active and silent system states, due to the dynamic nature of the system (Fig. 5c).

The evaluation of modifications on whole- and half-nucleosomes allows comparison of the model predictions with experimental observations of bivalent modifications on the same or opposite H3 tails[32,34,52]. In[52] and[32], H3K4me3 and H3K27me3 were not detected on the same histone tail by mass spectrometry. Using single molecule imaging[34] detected H3K4me3 on 0.5% of H3K27me3 tails. The reported co-occurrence of H3K36me2 with H3K27me3 is higher. In[52] 3.9% of H3K27me3 tails also carry H3K36me2. Our model predicts that the percentage of total H3K37me tails that are also methylated on H3K4/36 depends on

system state. In the silent system state, (conditions as in Fig. 3c), 9% of all H3K27me tails are also methylated on H3K4/36. This prediction is consistent with the observation of frequent co-occurrence of H3K36me and H3K27me on the same tail, but much higher than the observed co-occurrence of H3K4me3 and H3K27me3. We do not separate H3K4 and H3K36 methylation in the model. The fusion of these two modifications is justified by the biochemical data available, and has benefits in reducing the complexity of the model: the cube shown in Fig. 1b would be a four- dimensional structure if H3K4 and H3K36 were separated. However the cost of this reductionism is seen when comparing the model predictions to experimental data that show different behaviours of the two modifications. Further refinement of the model to treat H3K4 and H3K36 methylation separately will be interesting in future to address this discrepancy.

Interestingly, two different bivalent subtypes strongly correlate with active or silent system states. Active bivalent nucleosomes, containing containing H2A ubiquitylation and H3K27 acetylation, preferentially co-occur as a subpopulation accompanying the active system state (Fig. 4g, Fig. 5, Supplementary Fig. 6). Similarly, silent bivalent nucleosomes containing both H3K4/K36me and H3K27me co-occur with the silent system state (Fig. 4, Fig. 5, Supplementary Fig. 6). We interpret this bias towards active or silent bivalent nucleosomes as a result of the dynamic bistable nature of the system. These active and silent bivalent nucleosomes are only a few modifications away from their monovalent counterparts (Fig. 2b), and thus the system readily occupies the nearby states even when in a stable dominant active or silent mode. Thus in each stable mode, although monovalent states dominate on average, each nucleosome in the system periodically alternates between monovalent modes and the closest bivalent modes (Fig. 5c). The recent development of in vivo sensors for chromatin modifications and particularly for bivalent chromatin[73] may offer a direct means of testing these predictions in living cells.

To what extent are these predictions consistent with existing experimental observations? Surprisingly, the predominant active bivalent form that we predict (co-occurrence of H3K27ac and H2Aub on the same nucleosome) has not been addressed experimentally to our knowledge. It would be highly informative to determine whether this form exists, and whether it does indeed preferentially occur with active states. In contrast, bivalent chromatin containing H3K4me3 and H3K27me3, which we predict to be a substantial background in silent chromatin has been extensively characterised in vertebrates, but rarely observed in *Drosophila*[74,75]. Several vertebrate studies have noted a correlation between K4/K27me bivalent chromatin and silent gene expression states in ESC cells and also in differentiated cell types, which is consistent with the model prediction[21–25,29,32–34,51].

Although the active (H3K27ac/H2Aub) form of bivalent chromatin that we predict here has not been studied, two recent studies have addressed heterogeneity within bivalent chromatin carrying H3K27me3 and H3K4me3[30,76]. Both studies identified bivalent chromatin carrying higher amounts of H3K4me3 than H3K27me3 associated with active loci, and the converse (higher H3K27me3 than H3K4me3) associated with silent loci. Consistent with these observations, our model does indeed predict that the ratio of H3K27me to H3K4me in the population of bivalent nucleosomes carrying both marks is reduced as the system switches from the silent to the active regime (Supplementary Fig. 7e). It would be highly interesting in future to apply the approaches of[30,76] to the H3K27ac/H2Aub form, which we predict to display the opposite behaviour (i.e. the proportion of H3K27ac to H2Aub is expected to decrease as the system becomes silent (Supplementary Fig. 7d).)

In conclusion, our work prompts a re-evaluation of the potential biological role of bivalent chromatin. The proposition that bivalent chromatin is poised, and is resolved to active or silent states upon differentiation was largely based on ChIP experiments performed with a single-antibody against each of these marks (H3K4me3 or H3K27me3)[21–25]. We propose instead that bivalent H3K4/K27me nucleosomes are a subpopulation of silent chromatin in a dynamic system (Fig. 5c), and that true poised chromatin is bistable. As shown in Fig. 5f, the loss of H3K4me or H3K27me marks upon differentiation observed in single ChIP experiments would be consistent with a resolution of poised bistable chromatin, as we propose here. Sequential or co-ChIP experiments would give different results if poised chromatin is indeed bistable as we propose (Fig. 5f). If the bistable model is correct, then both K4/K27me nucleosomes and Ub/Ac nucleosomes should exist in bistable chromatin due to the mix of active and silent states each containing the preferred background of one of these bivalent forms. Each of these forms should then become preferentially detectable at stable active or silent loci (Fig. 4g, j, 5f).

Finally, we propose that future research will profit from a shift in focus toward studying modification states that are currently poorly characterised or have not been studied at all. Research to date has been strongly biased towards silent chromatin states and bivalent states containing H3K27me3 (Fig. 5i, j see "Methods" section for full reference list). We propose that a consideration of the less popular states will prove very fruitful.

## Methods

**Model summary**. The model consists of an array of half-nucleosomes, each containing a single copy of histones H2A and H3 (H2B and H4 are not considered in the model). Each pair of half-nucleosomes constitutes a whole nucleosome. The size of the system is given by the parameter $N$, indicating the number of half-nucleosomes. Thus if $N = 20$, there are 10 whole nucleosomes, representing approximately 2kb of DNA. Half-nucleosome 1 is selected at random from the array. Specific modifications on half-nucleosome 1 can recruit readers. Half-nucleosome 2 is selected at random. A reader recruited by half-nucleosome 1 can add a single modification to half-nucleosome 2, changing its state by one step. Rates of recruitment and modification are the same for all half-nucleosomes: halves that belong to the same nucleosome and those that belong to different nucleosomes obey the same rules, and the linear distance between nucleosomes is not taken into account. The system state is evaluated in terms of whole nucleosomes, which are categorized as shown in Fig. 2b.

**Histone modifications, writers and erasers**. The model contains four key histone modifications or groups of modifications that are important for PcG/TrxG regulation. The PcG mediated modifications in the model are histone H3 lysine 27 trimethylation (H3K27me3) and histone H2A lysine 118 or 119 monoubiquitylation (H2Aub). The TrxG mediated modifications in the model are histone H3 lysine 4 and/ or 36 methylation (H3K4/36me) and histone H3 lysine 27 acetylation (H3K27ac). Acetylation and methylation on H3K27 are mutually exclusive. For each of these four modifications, the enzymes or complexes responsible for its addition and for its removal have been identified and are included in the model, giving a total of eight catalytic conversions (Table 1). All of the enzymes and modifications, and most of the feedbacks and inhibitions included in the model have been documented genetically and/ or biochemically for both *Drosophila* and vertebrates, thus the model is potentially applicable to both. The model includes several simplifications for each modification, which are illustrated in Fig. 1c and outlined below.

H3K27me3: trimethylation of H3K27 is associated with gene silencing and is catalyzed by the Polycomb repressive complex 2 (PRC2) by the catalytic subunit EZH2 in vertebrates[77,78], and in *Drosophila* by the catalytic subunit E(Z)[79,80]. In addition, there exist mono- and dimethylated states of H3K27, which potentially have different roles in gene regulation and are also catalyzed by PRC2[81], but are not included in the model, which considers only the trimethylated state of H3K27 (Fig. 1b, d, half-nucleosome states 9–12). In the model the transition from a histone H3 that is unmodified on K27 to one that carries K27me3 is represented by PRC2 (Fig. 1d).

Demethylation of di and trimethylated H3K27me3 is catalyzed by the histone demethylases UTX (KDM6A) and JMJD3 (KDM6B) in vertebrates[46,82–84] and by dUTX in *Drosophila*[85], for review see ref. [86]. In vertebrates the dual demethylase KDM7 demethylates monomethylated H3K27 and H3K9[87]. In the model a single transition from trimethylated H3K27me3 to unmodified H3K27 is represented by UTX (Fig. 1d).

H2Aub: monoubiquitylated Histone H2A is associated with gene silencing and is catalyzed by the Polycomb repressive complex 1 (PRC1). Histone H2A is monoubiquitylated on lysine 118 in *Drosophila* by the PRC1 subunit dRING[88] and on lysine 119 in vertebrates by the PRC1 subunits RING1A and RING1B[88–90]. In both flies and vertebrates, non- canonical PRC1 complexes exist that contain the RING ubiquitin ligase but lack the Polycomb (*Drosophila*) or CBX proteins (vertebrates)[91,92] reviewed in[17]. In the model the transition from histone H2A in which lysine 119 or 118 is unmodified to one carrying a ubiquitin modification at this site is represented by a single reaction catalysed by PRC1 (Fig. 1d, half-nucleosome states 2,3,6,7,10 and 11).

De- ubiquitylation of H2Aub K118/119 is catalysed in both flies and vertebrates by the BAP-1 subunit of the PR-DUB complex (fly[93], vertebrate[94]). PR-DUB removes monoubiquitylation from H2AK118 or 119 but not from H2AK13/15[95] nor from H2B[93], thus it is specific for PRC1 mediated monoubiquitylation. No other enzymes that catalyse this reaction have so far been described. The de-ubiquitylation of H2Aub is represented in the model by the PR-DUB mediated transitions (Fig. 1d).

H3K27ac: acetylation of histone H3 on lysine 27 is associated with gene activation. The histone acetyltransferase CBP catalyses the acetylation of H3K27 and other residues in *Drosophila*[96]. In vertebrates, both CBP and p300 catalyse this reaction[40]. In the model the transition from a histone H3 tail in which K27 is unmodified to one in which K27 is acetylated is represented by a single reaction catalysed by CBP (Fig. 1d, half-nucleosome states 1-4).

The deacetylation of H3K27 (among many other residues) is catalysed by the NuRD (nucleosome remodelling and histone deacetylase) complex in vertebrates[97] and by the histone deacetylase RPD3 in *Drosophila*[96]. In the model the deacetylation of H3K27 is represented by a single reaction, catalysed by NuRD, giving a histone H3 tail that is unmodified at K27 (Fig. 1d).

H3K4/K36me: methylation of histone H3 on lysines 4 and 36 is associated with gene activation. The specificity of TRX and its vertebrate homolog MLL1 is disputed. Both have previously been reported to methylate H3K4me3[98–100]. However it was later shown that Most H3K4me3 is not catalysed by MLL1 in vertebrates[101,102] and TRX in *Drosophila*[103,104] but by the SET1/COMPASS complex. Two recent publications disagree on the specificity of TRX and MLL1. The first[42] reports that both catalyse H3K4me1 only. The second[105] reports that both TRX and MLL1 are specific for H3K4me2. Some of these discrepancies may be attributable to anti H3K4me antibody specificity[106]. However, for the purposes of our model, we are interested in any H3K4 methylation that genetically or biochemically antagonises PcG proteins, thus in the absence of a consensus and in the interests of simplification we refer to H3K4me.

*Drosophila* ASH1 and vertebrate ASH1L dimethylate H3K36. (Fly[43,107,108], vertebrate[109,110]. Both H3K4me and H3K36me are associated with active genes, and fly and vertebrate ASH1 (ASH1L in vertebrates) and TRX (MLL1) show highly similar localisation on chromatin at many loci and interact directly with each other (in fly:[37], in vertebrates:[38,39]). TRX binding to chromatin is dependent on ASH1 at flies[36] and on H3K36me2 at several loci in vertebrates[39]. For simplicity in the model the methylated H3K4 and H3K36 were fused to a single species (H3K4/36), (see Fig. 1d, half-nucleosome states 1, 2, 5, 6, 9 and 10). Likewise the activities of ASH1 and TRX were fused in the model to a single activity (named TRXG in Fig. 1d). We note that this simplification has costs, as situations in which the two proteins act independently of each other do exist[29,59]. However the cost to the model of considering them separately would be large: The 3 dimensional model of Fig. 1b would become four-dimensional, with a corresponding increase in unknown parameters. Furthermore any version of such a model that includes the interdependency of ASH1 and TRX would behave identically to one in which they are fused. Thus, at present, we consider that fusing the two is justified for the purposes of modelling but we do not wish to imply by this that they act in tandem in all possible situations.

Several demethylases act specifically on methylated H3K4 and H3K36 (reviewed in ref. [86]). For example JARID1A (synonyms: RBP2, KDM5A) demethylates H3K4me3 and me2 in vertebrates[86,111]. The *Drosophila* KDM5 homolog Lid (Little imaginal discs) demethylates H3K4me3 and me2 in flies[112,113]. Fbxl10 (synonyms: KDM2B, JHDM1A) demethylates H3K36me1 and me2 in vertebrates[114] and *Drosophila*[91]. KDM2B is also reported to be a H3K4me3 demethylase in flies[115] and in vertebrates[86]. In the model, the demethylation of H3K4 and K36 is represented by a single reaction, catalysed by KDM, giving a histone H3 tail that is unmodified at K36 and K4 (Fig. 1d).

The unmodified state: the half-nucleosome state designated as unmodified in the model (state 8, Fig. 1d) is unmodified on histone H3K4, K36 and K27, and on histone H2AK118 (flies) or K119 (vertebrates). Other residues that do not affect PcG/TrxG regulation are not relevant in the model and are not considered. The model makes the assumption that half-nucleosomes that are incorporated into newly replicated chromatin are in state 8 (see also model implementation below).

**Model structure**. Nucleosome states and transitions: the model contains 12 half-nucleosome modification states, comprising the unmodified state plus all possible combinations of the four modifications except the simultaneous presence of H3K27me3 and H3K27ac, which are mutually exclusive on the same histone tail[40–42]. The states are arranged in the model in a three-dimensional three- layered structure in which each layer contains four states (Fig. 1a). Each horizontal transition (within a layer) or vertical transition (between layers) changes the state of a

half-nucleosome by one modification (Fig. 1b, d). Forward and reverse transitions are catalyzed by the complexes described above, with the additional constraints for recruitment and inhibition of specific complexes described below (see also Tables 1–3). Half- nucleosomes are paired to make whole nucleosomes for the purposes of evaluation (Fig. 2).

Histone-modification dependent recruited conversions: numerous studies have reported histone modifications that can enhance binding to chromatin or stimulate the activity of specific PcG or TrxG complexes. These molecular interactions are included in the model as follows: a complex that can bind to a given histone modification acts as a reader of that modification on half-nucleosome 1, and can in turn recruit (write) to half-nucleosome 2 (Fig. 1c). These interactions are referred to as recruited conversions in the model. Names of complexes that are recruited to each of the 12 half-nucleosome states in the model are shown above the half-nucleosome state containing that modification in Fig. 1d.

The following states can enhance binding and/ or stimulate specific complexes and are implemented as recruitments in the model:

- H3K27me3 can bind PRC1 (Fly:[116–118], vertebrate:[119]). In the model half-nucleosome states 9–12 recruit PRC1 (Fig. 1d).
- H3K27me3 can bind vertebrate PRC2[120,121]. Binding occurs via the EED subunit of PRC2 and stimulates PRC2 methyltransferase activity[121]. Genetic evidence for a similar mechanism in *Drosophila* was provided in[121]. In the model, half-nucleosome states 11 and 12 recruit PRC2 (Fig. 1d).
- H2Aub can bind PRC2 (Vertebrate:[122–125], fly[122]). Binding of vertebrate PRC2 to H2Aub stimulates PRC2 methyltransferase activity in vitro[122]). In the model, half-nucleosome states 3, 7 and 11 recruit PRC2 (Fig. 1d).
- H3K4 monomethylation stimulates the activity of *Drosophila* CBP in acetylating H3K27[42,96]. In the model, half-nucleosome states 1,2,5 and 6 recruit CBP or TRX:CBP via this interaction (Fig. 1d). Although half-nucleosome states 9 and 10 also contain H3K4me, they do not recruit CBP due to inhibition.
- H3K27ac enhances binding of TrxG proteins to chromatin. The vertebrate TrxG protein BRD4 (fly homolog: FSH(1)) binds acetylated histones[126]. *Drosophila* FSH binds acetylated histones, interacts physically with ASH1 and may recruit or stabilise ASH1 at some loci[127,128]. Since ASH1 and TRX interact directly with each other (in fly: ref. [37], in vertebrates: ref. [38,39]), we simplify these interactions in the model so that both are represented by TRXG. Half-nucleosome states 1-4 contain H3K27ac and recruit TRXG and/or TRXG:UTX and TRXG:CBP.

Histone modification-independent conversions: there is ample evidence that not all recruitment of PcG and TrxG proteins depends on pre-existing histone modifications. Documented mechanisms of histone – independent recruitment include specific DNA sequences, non- specific and specific DNA binding proteins, demethylated DNA, non-coding RNAs, and RNA polymerase (reviewed in refs. [17,18]. For example, both PRC2[49,50] and PRC1[47,129] bind DNA non- specifically with higher affinity than for specific histone tail modifications.

In addition, histone loss due to turnover, transcription and replication leads to loss of modifications. Replication is explicitly modelled (see Model implementation below). For all other histone modification-independent recruitments or removals, the term 'beta', applied to all transitions, describes the rate of conversion between states that is independent of recruitment. For simplicity we use a single value of beta for all histone modification-independent transitions. If more quantitative information becomes available in future, beta can be adjusted separately for specific transitions, for example all those catalyzed by PRC1, to reflect the recruitment of canonical and non- canonical PRC1 to chromatin independently of H3K27me3[118,130–132] (reviewed in[17]).

Histone modification-mediated inhibition: recently, several studies have identified histone modifications that inhibit the activity of specific PcG or TrxG complexes. These inhibitions are formalised in the model and simplified in some cases, as described below.

- Histone H2A ubiquitination inhibits the enzymatic activity of H3 lysine 36 methyltransferases[45]. This is represented in the model in two ways. Firstly, by the fact that half-nucleosome states 2 and 3, containing H2Aub, do not recruit TRXG, despite the presence of H3K27ac, which would normally cause recruitment (Fig. 1c). In the model, this means that a half-nucleosome in state 2 or 3 will not be able to recruit TRXG and will not be able to add H3K4/ K36me to a second half-nucleosome. Secondly, if the first half-nucleosome can recruit TRXG (ie it has H3K27ac but not H2Aub; state 1 or 4), but the second half-nucleosome chosen does contain H2AUb, then addition of H3K4/K36me to this second half-nucleosome by TRXG is prevented. In the model, this affects the conversion of state 11 to 10, state 7 to 6 and state 3 to 2. These transitions are governed only by beta, describing the rate of non-recruited conversions (indicated by grey arrows on Fig. 1d).
- H3K4 and H3K36 methylation inhibit PRC2 H3K27 methylation activity (*Drosophila* PRC2[43,44], vertebrate PRC2[32,44]. This is represented in the model in two ways as described above. Firstly, half-nucleosome states 9 and 10 contain H3K4/36me and thus do not recruit PRC2 despite the presence of modifications that would normally recruit PRC2 (H3K27me3 in both states, and H2Aub in state 10; Fig. 1d). Secondly, any PRC2- mediated transition of the second half-nucleosome towards H3K27me3 is prevented if that second

half-nucleosome contains H3K4/K36me. In the model, this affects the conversion of state 5 to 9 and state 6 to 10, which are governed only by beta (indicated by grey arrows on Fig. 1d).

- PRC2 activity is inhibited by nucleosomes containing H3K4me3 or H3K36me3 on both tails (symmetric), but not if only one tail is modified (asymmetric)[32]. This is represented in the model by the relationship between half- and whole nucleosomes (Fig. 2b). A half-nucleosome that already carries H3K4/K36me is not a substrate for PRC2 as explained above. This means that a nucleosome in which both halves carry H3K4/K36me (symmetric) will not be modified by PRC2. In contrast, the presence of H3K4/K36me on only one H3 tail of a nucleosome (asymmetric) does not prevent modification of the other H3 tail of that nucleosome.
- Polycomb (subunit of PRC1) inhibits histone acetylation mediated by CBP by binding directly to the CBP catalytic domain[133]. In the model, half-nucleosome states 9 and 10 do not recruit CBP despite the presence of H3K4me, because they recruit PRC1 (Fig. 1d).
- Human SET1 and MLL1 complexes bind poorly to H3K27me3 histones. Catalytic activity is not prevented[134]. In the model, half- nucleosome states 9–12 do not recruit TRXG (Fig. 1d). However, if the first half-nucleosome can recruit TRXG, but the second half-nucleosome contains H3K27me3 and not H2Aub (state 12), then addition of H3K4/K36me to this second half-nucleosome by TRXG is allowed.

Compound complexes. Several complexes or proteins have been found in direct association with each other, and are formally represented as separate compound species in the model, indicated above the states, with double names, e.g., PRC2:KDM. This feature of the model enables recruitment and activity to be separated in specific cases. For example, the enzymatic activity of PRC2 in methylating H3K27, but not its binding to H3K27me3, is inhibited by the presence of H3K4 and H3K36 methylation on nucleosomal substrates[32,43,44]. This is implemented in the model as follows: a PRC2 complex that attempts to bind to a half-nucleosome in state 9 or 10 (which have both H3K27me3 and H3K4/36me) will be able to bind via the H3K27me3 modification, but will not be able to modify another half-nucleosome in the array, because its activity is inhibited by H3K4/K36me. The binding of PRC2 would however, recruit other enzymes that are associated with PRC2, and whose activity is not inhibited by the H3K4/K36 modification. An example is the PRC2:KDM dual complex (see below for more detail). In the model, the PRC2 part of PRC2:KDM can recruit to a first half-nucleosome, and the KDM part can demethylate H3K4/K36 on the second half-nucleosome chosen from the array, if it is present. The following compound complexes are represented in the model, following similar logic: (N.B. for each compound complex in the model, only the last enzyme listed has catalytic activity)

- PRC2:KDM. Vertebrate PRC2 recruits RBP2 (H3K4 demethylase)[111] and LSD1 (H3K4 demethylase)[135]. Recruited via H3K27me3 and H2Aub to all states that contain these modifications (half-nucleosome states 2, 3, 6, 7, 9–12, (Fig. 1d)).
- PRC2:KDM:NURD. The inclusion of this complex in the model is based on the above observations that PRC2 associates with H3K4/K36 lysine demethylases, and that several H3K4/K36 lysine demethylases in turn associate with the deacetylases NuRD (vertebrates) or RPD3 (*Drosophila*). For example, the *Drosophila* H3K4 demethylase Lid interacts physically and functionally with RPD3[136]. The vertebrate H3K4/K6 demethylase LSD1 is a component of the NuRD complex at active enhancers[137]. These interactions are summarized in the model by the compound complex PRC2:KDM:NURD, which is recruited via PRC2 binding to H3K27me3 or H2Aub, to all states containing one or both of these modifications (half-nucleosome states 2, 3, 6, 7, 9–12; Fig. 1d).
- TRXG:CBP. ASH1 and CBP interact physically and functionally[138]. The TrxG protein BRM is associated with CBP and stimulates the activity of CBP in acetylating H3K27[139]. These interactions are represented in the model by the compound complex TRXG:CBP (Fig. 1d).
- TRXG:UTX. Vertebrate UTX (H3K27 demethylase) is associated with MLL 2/3 (vertebrate homologs of TRX)[46]. *Drosophila* UTX (H3K27 demethylase) is associated with CBP and the TrxG protein BRM[139]. These interactions are represented in the model by the compound complex TRXG:UTX (Fig. 1d).
- PRC1:KDM. Not explicitly represented in the model are the non-canonical PRC1-KDM complexes dRAF and PRC1.1, which contain the ubiquitin ligase dRING (flies) or RING1A/ RING1B (vertebrates) and H3K36 demethylase dKDM2 (flies[91]) or KDM2B (vertebrates[92]). These complexes lack the chromodomain containing subunits PC (flies) or CBX (vertebrates) and thus are not recruited to chromatin via the interaction between the PC chromodomain and H3K27me3 (for reviews, see[17,140,141]). No recruitment via existing histone modifications has yet been reported for non-canonical PRC1, thus at present these complexes are covered by the recruitment-independent parameter beta.

**Model implementation.** The model is implemented as an agent based model on $N = 20$ units, that each can be in any of the 12 half-nucleosome states shown in

Fig. 1b, d. The model lets these units interact and modify each other as specified by seven read - write or read - erase enzymes (1: TRX; 2: UTX = TRX:UTX; 3: CBP = TRX:CBP; 4: PRC1; 5: PRC2; 6: PRC2:KDM; 7: PRC2:KDM:NURD), each of which has one or zero ways to modify each of the 12 allowed states.

The update uses an event-based Gillespie type algorithm. At each update step one selects the next time for potential occurrence of one of four types of moves:

1. A recruitment type process with rate equal to one, set to occur at time increment t1=-ln(ran1).
2. Direct conversion with rate beta, t2=-ln(ran2)/beta.
3. A direct acetylation move with rate NURD (dn), t3=-ln(ran3)/dn.
4. A direct de-ubiquination move with rate PR-DUB (dp), t4=-ln(ran4)/dp.

In these, time steps ran1, ran2, ran3 and ran4 are four different random numbers that each are selected from uniform distribution between 0 and 1. The next attempted move is pinpointed by the move type with the smallest time increment among t1, t2, t3 and t4.

- Recruitment move (1, i.e., t1 smallest): In case that a recruitment move is selected (t1 smallest) then a recruiting half-nucleosome is chosen randomly among the N half- nucleosomes. The state of this half-nucleosome specifies which recruitment processes are allowed to act. In case the recruiting half-nucleosome is in state 8 there is no recruitment possible and no further action is taken for this update step. Otherwise one selects randomly, with equal weight, one of the allowed read-write or read- erase enzymes that can be recruited by the state of the recruiting half-nucleosome. Subsequently a target half-nucleosome is selected at random. If the selected read-write enzyme can modify the state of the target, then this target half-nucleosome changes state as specified by the write or erase processes on the directed arrows in Fig. 1d. If the enzyme cannot change the state of the target half-nucleosome then no further action is taken for this update step. Otherwise the target half-nucleosome is assigned the new state specified by the enzyme. Notice that each read-write or read- erase enzyme uniquely specifies which change it can make to any of the 12 half-nucleosome states in the model (Figure 1d).

- Random direct move (2 i.e. t2 smallest): Select a random half-nucleosome, and select with equal chance which of the modification positions (H3K27, K4/36, H2A) should be changed.

If position H3K27 is selected, evaluate the state of the modification on this position of the half-nucleosome. If this state is H3K27ac or H3K27me then it is changed to the non-modified state of H3K27. If H3K27 is unmodified then it is changed with equal probability to either an acetylated or to a methylated state. If position K4/36 is selected, reverse the present state of the half-nucleosome between the methylated and the non-methylated state. If position H2A is selected, reverse the present state of the half-nucleosome between the ubiquitinated and the non-ubiquitinated state.

- Direct move due to non-recruited NURD activity (3, i.e., t3 smallest): Select a random half-nucleosome, and if this has modification H3K27ac, then its state is changed to non-modified H3K27.

- Direct move due to non-recruited PR-DUB activity (4, i.e. t4 smallest): Select a random half-nucleosome, and if this has modification H2Aub, then its state is changed to non-modified H2A.

Notice that change is only executed when the above conditions are met, and that there will be situations where the attempted move is not successful. In any case the overall time-counter is updated by adding a time increment = min (t1,t2,t3,t4). Thus in each update step, a maximum of one of the above changes is executed. Subsequently the next update step is started by again generating ran1- ran4.

In simulations, we also consider replication, when accumulated time since last replication reaches a generation time of 20 multiplied by the system size N. At replication we select each half-nucleosome and with 50% probability replace it with one in the unmodified state 8. Thus on average 50% of the half-nucleosomes are replaced, but stochastic partitioning variations are allowed. In figures, we plot the dynamics in units of time counter divided by number of half-nucleosomes (i.e. number of recruitment attempts per half-nucleosome). In some simulations, we vary the strength of a subset of the recruitment reactions. When a recruitment reaction is lowered, this is done by accepting an otherwise acceptable move with a reduced probability. When a recruitment reaction is set to be larger than one, we select all recruitment reactions with larger rate, and reduce the acceptance of the non-selected ones. For evaluation of the output of simulations, half-nucleosomes are assigned to pairs according to their position in the array. Each whole nucleosome thus created is categorized according to Fig. 2b.

We emphasize that the above implementation is hugely simplified. In particular it employs a nucleosome-centered view of recruitment, where each half-nucleosome selects each of its read-write or read- erase enzymes with equal rate (set equal to one). This will cause selection of different read-write enzymes to occur with different frequency. There is no reason to refine this approach

before we have information about actual recruitment rates. At present, the proof-of-concept model that we present here represents a baseline of current knowledge for exploring systemic features of nucleosome modification dynamics.

**Citation density landscape**. The citation density landscape shown in Fig. 5i gives an indication of the popularity of each of the 12 states considered in our model, as indicated by citation frequency of papers relating to each state. The landscape plots -log(total citations) for each state. The total citations for each state were calculated for the 73 publications used to formulate the model[32,37,38,40–47,49,50,77–100,105,107–109,111–125,127–139,142], and 17 publications relating to bivalent chromatin[21–27,29–34,51,75,76,143]. The bivalent chromatin publications were selected to represent the first reports of bivalent chromatin in mouse and human ESCs[21–24,26,27], and in differentiating systems[25], but do not cover publications on iPSCs, since these are highly cited for reasons other than bivalent chromatin itself. Some papers also report bivalent chromatin containing H2AUb[26,27,31]. We found a single publication addressing H3K4/K27me bivalent chromatin in *Drosophila*[75]. Finally, we include recent re-evaluations based on mass spec[32], imaging[34], and co-ChIP or sequential ChIP[29,30,33,51], demonstrating the co- occurrence of opposite modifications on the same or adjacent nucleosomes. The number of citations for each reference was calculated according to Google Scholar on 26.01.2019.

To generate the citation density landscape, total citations were added to each of the 12 half-nucleosome states (Fig. 1d) according to the following rules:

- Addition of a modification: Citations of papers demonstrating addition of a modification are added to all half-nucleosome states that carry the modification.
- Removal of a modification: Citations are added to the half-nucleosome state from which the modification is removed.
- Exclusivity of H3K27ac and H3K27me3: Citations of papers documenting this mutually exclusive relationship are added to all states that contain either H3K27me3 or H3K27ac, and to the unmodified state 8.
- PRC1 can bind independently of H3K27me3: Citations are added to all H2Aub states (ie 2, 3, 6, 7, 10, 11).
- Recruitment: Citations are added to the state that recruits, and to the state containing the modification that is added or erased by the recruited enzyme.
- Inhibition: Citations are added to the state that inhibits, and to those containing the modification that is inhibited.
- Bivalent chromatin: Citations are added to the bivalent half-nucleosome states that are addressed by the paper (yellow dots on Fig. 5i).

This analysis generated the following total citations for each state:

| State | Citations |
|---|---|
| 1 | 6344 |
| 2 | 17261 |
| 3 | 10671 |
| 4 | 1415 |
| 5 | 5232 |
| 6 | 15808 |
| 7 | 9176 |
| 8 | 1391 |
| 9 | 33089 |
| 10 | 28568 |
| 11 | 22232 |
| 12 | 15988 |

**Reporting Summary**. Further information on research design is available in the Nature Research Reporting Summary linked to this article.

## Data availability

All relevant data supporting the key findings of this study are available within the article and its Supplementary Information files or from the corresponding author upon reasonable request. A reporting summary for this Article is available as a Supplementary Information file.

## Code availability

All computer code used in this study is available on request.

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

## Acknowledgements

L.R. thanks Marc Rehmsmeier for discussions. KS acknowledges support by the Danish National Research Foundation through the Center for Models of Life, and by the European Research Council under the European Union's Seventh Framework Programme (FP/2007 2013)/ERC Grant Agreement n. 740704. L.R. acknowledges the support of the German National Research Foundation (DFG) Exzellenzinitiativ IRI-Lifesciences https://www.iri-ls.hu-berlin.de/en and of the EU MSCA ITN 'PEP-NET' Grant Agreement n. 813282.

## Author contributions

Both authors contributed to model design, conceptual project design and writing the manuscript. L.R. performed literature curation. K.S. performed modelling.

## Additional information

**Competing interests:** The authors declare no competing interests.

