## [Peer Review File · Nature Communications]

Reviewers' comments:

Reviewer #1 (Remarks to the Author):

In their manuscript 'Comprehensive theoretical analysis of the Polycomb – Trithorax system predicts that poised chromatin is robustly bistable' Sneppen and Ringrose develop a mathematical modeling framework to explore the relationship between active, silent, and poised states set up by the Trithorax and Polycomb systems of chromatin modifiers. Taking into account 64 publications on the biochemistry of the activities involved, their model predicts the system to be inherently bistable with minimal occurrence of so-called 'bivalent' states, including the H3K4me3-H3K27me3 bivalent state that has been described in embryonic stem cells and other systems. The authors propose that poising is mediated by a rapid switching between active and repressed states rather than via a distinct bivalent state.

The findings of the manuscript are interesting and helpful to the field, adding a new angle to the ongoing debate around bivalent chromatin and its functions. The general concept of overall bistability of the trxG–PcG system has been proposed in one form or another by several members of the field before (see e.g. Mendenhall et al. PLoS Genetics 2010, Klose et al. PLoS Genetics 2013) and is now largely accepted, however, invoking this concept has so far been based mostly on an intuitive understanding of the system rather than a rigorous mathematical framework. As such, the present study could fill an important gap.

However, with respect to bivalent chromatin, the study leaves out important experimental evidence that is likely to significantly alter the predictions of the model and their interpretation. Most crucially, it has been shown by both the Reinberg (Voigt et al. Cell 2012) and Bernstein (Shema et al. Science 2016) labs that bivalent nucleosomes are in an asymmetric configuration where the H3K4me3 and H3K27me3 marks occupy separate histones within the same nucleosome. The model used in the present manuscript considers half nucleosomes as the relevant unit and therefore misses these asymmetric states and their consequences. These papers, as well as others, also show that H3K4me3 and H3K27me3 do not co-occur on the same histone. The prediction of the model that H3K4me3-H3K27me3 states do not occur on half-nucleosomes is therefore not quite as surprising as the authors suggest. The model would need to be adapted to consider full nucleosomes in order to make valid and meaningful predictions that correspond to what has been observed experimentally. At present, the conclusions drawn from the model are based on partially inaccurate assumptions, rendering their validity and usefulness questionable.

A manuscript that addresses this central point along with additional issues outlined below would make a valuable contribution to the field that could be well suited for Nature Communications.

Detailed points:

1.) As discussed above, the simplification in the model to define bivalency on the level of half nucleosomes does not agree with experimental evidence. The model should be altered to reflect the observation that H3K4me3 and H3K27me3 occur on separate copies of H3 within the nucleosome, requiring consideration of full nucleosomes rather than half nucleosomes to capture their occurrence. Unless this is done, the model will not be able to capture what has been shown experimentally.

Altering the model in that way might significantly change its predictions, making it difficult to comment on large parts of the manuscript in its current form. Unless the major outcome would again be prevalence of bistability rather than bivalency, the main conclusions of the manuscript would need to be altered and discussed accordingly.

It is currently challenging to determine the fraction of bivalent as opposed to admixed monovalent nucleosomes at a given promoter in a population of cells. The predictions of the model with regard

to the extent of bivalency are therefore difficult to verify experimentally. The authors might want to consider discussing the work of the Ruthenburg lab (Grzybowski et al. Mol Cell 2015) reporting calibrated ChIP-seq data giving modification densities as well as recent genome-wide reChIP work from several groups (see e.g. Kinkley et al Nat Commun 2016) when comparing predictions of the model to experimental data with regard to the extent of bivalency.

Given that bivalency has been observed experimentally, it would be helpful to use the revised model to identify situations, i.e. sets of parameters, where stable bivalency can be observed. If not a general feature of the system, could for instance different ratios of PcG to trxG activities explain the observation of bivalency? These might be useful predictions as well, as they could be tested experimentally. Along similar lines, parameters could be identified where stable transmission of a bivalent state through replication were possible. Again, this might lead to experimentally testable predictions, adding value to the model.

2.) Another simplification made in the model is to treat H3K4me3 and H3K36me3 as identical for the purpose of this model. Based on the observation that several mammalian H3K36 methyltransferases (or their SET domains rather than the full complexes) are inhibited by H2Aub, the authors state that trxG enzymes are inhibited by H2Aub. This is a potentially misleading generalization to make. To my knowledge, there is currently no evidence that H2Aub inhibits H3K4 methyltransferases. Given that H3K4 and H3K36 methyltransferases are very different in their architecture and regulation, the authors might want to reconsider this simplification. Other potentially very significant differences between H3K4 and H3K36 systems concern recruitment of the methyltransferases and removal of the marks. Generalizations about these aspects of the pathways are likely to be misleading as well. Moreover, the H2Aub-mediated inhibition has only been shown for mammalian enzymes, making the generalization towards *Drosophila* somewhat tenuous.

The model would be equally appealing but based on stronger foundations if H3K36 was taken out of the equation. As bivalent domains occur at promoters, the authors should limit the model to modifications that occur at promoters and exclude H3K36 methylation.

3.) It would be helpful to present the references that the different interactions, feedback loops, and inhibitory relationships in the model are based on in a table, giving references for both mammalian and *Drosophila* pathways in separate columns.

Demonstration of an interaction between Ash1L and MLL1 is being ascribed to Gregory et al. MCB 2007. However, this paper does not present any biochemical interaction data for these complexes and also states that Ash1L occupancy on chromatin is independent of MLL1.

In general, care should be taken when discussing data on MLL methyltransferases. These have undergone several rounds of renaming, most notably MLL4 having been referred to as MLL2 previously. The authors should make sure that findings from older references are linked to the correct gene/protein. This will should resolve some of the discrepancies discussed in Methods section 2.4. It is not clear why SET1 complexes in mammals were not mentioned.

Reviewer #2 (Remarks to the Author):

Reviewer Comments for "Comprehensive theoretical analysis of the Polycomb – Trithorax system predicts that poised chromatin is robustly bistable and minimally bivalent"

This manuscript synthesizes a wealth of molecular and biological information about the PRC-TRX chromatin system into a model for predicting the distributions and dynamics of chromatin modifications at a small generic locus. The major claim of the paper is that chromatin states containing nucleosomes with both active and silent modifications (bivalent) are expected to be rare. Instead, this particular model predicts that chromatin states that switch back and forth between containing nucleosomes with fully active or fully silent modifications are to be expected. The model also predicts novel bivalent states that were not yet tested experimentally. The subject of the manuscript is very timely. The chromatin/epigenetics field has been collecting a lot of data over the last few decades, and it is therefore ripe for complex models. This type of molecular simulations can nicely complement other efforts in the directions (such as machine learning on genome-wide data) in order to extracting general principles. Moreover, we appreciate the push the authors make for shifting the attention of the field from static snapshots of chromatin modifications to methods that could measure the dynamics of these states. This shift is especially needed for understanding poised genes, which by definition are genes where the dynamics of activation or silencing are supposed to be different.

The authors do an amazing job curating and summarizing the literature! That in itself is a big contribution. They also describe their model assumptions and parameters well, and actually make testable predictions.

While overall enthusiastic about the manuscript, we have some concerns about the assumptions that went into the model. Comparing the predictions of this model with other competing models in the field would greatly strengthen the manuscript. In addition, we have a few questions/suggestions about the definitions of terms in the paper and about choices made in the simulation parameters.

Major comments:

1. The authors assume a very strong positive feedback between chromatin modifications: chromatin regulators bound at a single modified nucleosome can modify all the other nucleosomes in the array equally. This feedback contributes greatly to the dynamics bistability they predict. However, it is not clear how much experimental evidence there is to justify this strong feedback assumption. In fact, other models in the literature assume that the feedback only influences nucleosomes that are adjacent to the modified one (or that the strength of feedback decreases with distance):

-Berry 2017 (ref 11 in this manuscript),

-Hathaway et al., Dynamics and Memory of Heterochromatin in Living Cells, Cell (2012)

-Hodges et al., Dynamics of inherently bounded histone modification domains, PNAS (2012)

-Erdel et al, Generalized nucleation and looping model for epigenetic memory of histone modifications, PNAS (2016)

-see also Erdel, How Communication Between Nucleosomes Enables Spreading and Epigenetic Memory of Histone Modifications, BioEssays 2017 for a discussion on new experimental evidence for weak feedback

Yes, it is true that some of the references above model HP1/H3K9 methylation. But the math is the same. And for either modification the feedback could be local, global (as assumed here), or in between.

It would be useful if the authors tried the other extreme assumption (feedback strictly between adjacent nucleosomes), and showed how their predictions change. This way, experimentalists can follow up and can actually discard one model or the other.

2. The definition the authors choose for bivalent nucleosomes is very restrictive, and might artificially make it seem like they are very rare. The authors define bivalent nucleosomes using half nucleosomes: the opposing modifications have to be on the same half of the nucleosome to call it bivalent. This is not the standard definition, it is not what is usually measured experimentally (see Shema, 2016, ref 33 and Weiner 2016, ref 28), and it might not be what matters for effectors biologically. More commonly, bivalent nucleosomes could have the opposing modifications on the

same half or on opposing halves of the same nucleosome. How common are these type of bivalent nucleosomes with modifications on the same nucleosome (on either side) in the results of the simulations? If they are quite common, it could change the conclusions of the paper.

3. I believe it would improve the paper to make the definition of poised clearer and more explicit from the very beginning (line 8 on page 4 for example). Especially given that unfortunately, in the literature often poised and bivalent are used interchangeably. Is poised supposed to imply the ability to rapidly switch from a low/off expression state to an on state upon stimulation (change in parameters)? The authors did not explore this.

4. The authors only simulate 10 nucleosomes. Is that enough? Domains of modified nucleosomes are at least an order of magnitude larger than that (15kb or larger, so 100 nucleosomes or larger). So are gene bodies in mammalian cells. How would the conclusions change if the authors simulated a larger nucleosomal array?

5. It would be interesting to see the single nucleosome trajectories in the simulations. Are bivalent nucleosomes common intermediates in the bistable switching?

6. It would improve the paper (and reproducibility) if some of the simulation details are better justified or documented. For instance:

a. how was the number of steps in the simulation chosen? Based on convergence?

b. In Supplementary Table 1, please label what the parameters in the columns represent (in a way that can easily be connected with the methods section, so that results can be reproduced).

Presumably these are rates.

7. How do the results of single-molecule Co-CHIP (Weiner 2016, ref 28), which analyzed a lot of combinations of modifications, compare with the predictions of the model?

Minor comments:

1. References to Figure 2D and 2E on page 13, lines 1 and 2 should be changed to Figure 3D and 3E, respectively.

2. On the following pages/lines references to Figure 1C should be changed to Figure 1D: pg 21 line 24; pg 22 lines 5, 14; pg 23 lines 5, 11, 22; pg 24 line 8; pg 25 lines 15, 17, 19, 21; pg 27 lines 7, 15, 23; pg 29 line 10.

3. The blue circles in Figure S2 can be difficult to distinguish from the black circles; perhaps changing the color, for example to cyan, would make this figure easier to interpret.

Reviewers' comments:**Reviewer #1 (Remarks to the Author):**

In their manuscript 'Comprehensive theoretical analysis of the Polycomb – Trithorax system predicts that poised chromatin is robustly bistable' Sneppen and Ringrose develop a mathematical modeling framework to explore the relationship between active, silent, and poised states set up by the Trithorax and Polycomb systems of chromatin modifiers. Taking into account 64 publications on the biochemistry of the activities involved, their model predicts the system to be inherently bistable with minimal occurrence of so-called 'bivalent' states, including the H3K4me3-H3K27me3 bivalent state that has been described in embryonic stem cells and other systems. The authors propose that poising is mediated by a rapid switching between active and repressed states rather than via a distinct bivalent state.

The findings of the manuscript are interesting and helpful to the field, adding a new angle to the ongoing debate around bivalent chromatin and its functions. The general concept of overall bistability of the trxG–PcG system has been proposed in one form or another by several members of the field before (see e.g. Mendenhall et al. PLoS Genetics 2010, Klose et al. PloS Genetics 2013) and is now largely accepted, however, invoking this concept has so far been based mostly on an intuitive understanding of the system rather than a rigorous mathematical framework. As such, the present study could fill an important gap.

Response:

We have cited Klose (page 3 line 20 ref 16), but we did not find any reference to bistability in Mendenhall.

However, with respect to bivalent chromatin, the study leaves out important experimental evidence that is likely to significantly alter the predictions of the model and their interpretation. Most crucially, it has been shown by both the Reinberg (Voigt et al. Cell 2012) and Bernstein (Shema et al. Science 2016) labs that bivalent nucleosomes are in an asymmetric configuration where the H3K4me3 and H3K27me3 marks occupy separate histones within the same nucleosome. The model used in the present manuscript considers half nucleosomes as the relevant unit and therefore misses these asymmetric states and their consequences. These papers, as well as others, also show that H3K4me3 and H3K27me3 do not co-occur on the same histone. The prediction of the model that H3K4me3-H3K27me3 states do not occur on half-nucleosomes is therefore not quite as surprising as the authors suggest. The model would need to be adapted to consider full nucleosomes in order to make valid and meaningful predictions that correspond to what has been observed experimentally. At present, the conclusions drawn from the model are based on partially inaccurate assumptions, rendering their validity and usefulness questionable.

A manuscript that addresses this central point along with additional issues outlined below would make a valuable contribution to the field that could be well suited for Nature Communications.

Detailed points:

1.)

(a)As discussed above, the simplification in the model to define bivalency on the level of half nucleosomes does not agree with experimental evidence. The model should be altered to reflect the observation that H3K4me3 and H3K27me3 occur on separate copies of H3 within the nucleosome, requiring consideration of full nucleosomes rather than half nucleosomes to capture their occurrence. Unless this is done, the model will not be able to capture what has been shown experimentally.

Altering the model in that way might significantly change its predictions, making it difficult to comment on large parts of the manuscript in its current form. Unless the major outcome would again be prevalence of bistability rather than bivalency, the main conclusions of the manuscript would need to be altered and discussed accordingly.

Response

We thank the reviewer for this excellent suggestion and we agree it makes total sense. We have now adapted the model to consider whole nucleosomes (Figure 2, Figure S1). In all subsequent figures, the system state is evaluated in terms of whole nucleosomes. The conclusions are very similar. Remarkably, despite the ability to adopt many more bivalent states, we find that the key conclusions of the original manuscript still hold true when considering whole nucleosomes. **The system remains robustly bistable and minimally bivalent.**

In particular regarding asymmetric nucleosomes: Methods, Page 33, line 16- 23. (Ref 32: Voigt et al., 2012).

“PRC2 activity is inhibited by nucleosomes containing H3K4me3 or H3K36me3 on both tails (symmetric), but not if only one tail is modified (asymmetric)³². This is represented in the model by the relationship between half- and whole nucleosomes (Fig. 2b). A half-nucleosome that already carries H3K4/K36me is not a substrate for PRC2 as explained above. This means that a nucleosome in which both halves carry H3K4/K36me (symmetric) will not be modified by PRC2. In contrast, the presence of H3K4/K36me on only one H3 tail of a nucleosome (asymmetric) does not prevent modification of the other H3 tail of that nucleosome.”

Results Page 21 (Ref. 32: Voigt et al., 2012, Ref: 34: Shema et al., 2016).

“The evaluation of modifications on whole- and half-nucleosomes allows comparison of the model predictions with experimental observations of bivalent modifications on the same or opposite H3 tails^{32,34}. In³², H3K4me3 and H3K27me were not detected on the same histone tail by mass spectrometry. The same study detected H3K36me2 on 25% of H3K27me2/3 tails. Using single molecule imaging,³⁴ detected H3K4me3 on 0.5% of H3K27me3 tails. Our model predicts that the percentage of total H3K37me tails that are also methylated on H3K4/36 depends on system state. In the active system state (conditions as in Fig. 3a), 45% of all H3K27me tails are also methylated on H3K4/36. In the silent system state, (conditions as in Fig. 3c), 9% of all H3K27me tails are also methylated on H3K4/36. These numbers are consistent with the observation of frequent co-occurrence of H3K36me and H3K27me on the same tail, but higher than the observed co-occurrence of H3K4me3 and H3K27me3.”

(see also response to point 2a below).

(b) It is currently challenging to determine the fraction of bivalent as opposed to admixed monovalent nucleosomes at a given promoter in a population of cells. The predictions of the model with regard to the extent of bivalency are therefore difficult to verify experimentally. The authors might want to consider discussing the work of the Ruthenburg lab (Grzybowski et al. Mol Cell 2015) reporting calibrated ChIP-seq data giving modification densities as well as recent genome-wide reChIP work from several groups (see e.g. Kinkley et al Nat Commun 2016) when comparing predictions of the model to experimental data with regard to the extent of bivalency.

Response:

We thank the reviewer for this excellent idea and for drawing our attention to these publications. We have done this comparison and present results in Figure S7 and text on page 23 (Results section). Ref. 30: Kinkley et al., 2016; Ref. 75: Grzybowski 2015.

“...two recent studies have addressed heterogeneity within bivalent chromatin carrying H3K27me3 and H3K4me3^{30,75}. Both studies identified bivalent chromatin carrying higher amounts of H3K4me3 than H3K27me3 associated with active loci, and the converse (higher H3K27me3 than H3K4me3) associated with silent loci. Consistent with these observations, our model does indeed predict that the ratio of H3K27me to H3K4me in the population of bivalent nucleosomes carrying both marks is reduced as the system switches from the silent to the active regime (Fig. S7e). It would be highly interesting in future to apply the approaches of^{30,75} to the

H3K27ac/H2Aub form, which we predict to display the opposite behaviour (i.e. the proportion of H3K27ac to H2Aub is expected to decrease as the system becomes silent (Fig. S7d).”

(c) Given that bivalency has been observed experimentally, it would be helpful to use the revised model to identify situations, i.e. sets of parameters, where stable bivalency can be observed. If not a general feature of the system, could for instance different ratios of PcG to trxG activities explain the observation of bivalency? These might be useful predictions as well, as they could be tested experimentally. Along similar lines, parameters could be identified where stable transmission of a bivalent state through replication were possible. Again, this might lead to experimentally testable predictions, adding value to the model.

Response:

We have performed parameter scans over several orders of magnitude for all pairs of parameters (Figure 3, Figures S3 and S4). We find that bivalent chromatin is difficult to maintain except under extreme conditions.

This is documented in Figure S5, and page 12 (Results):

“Indeed it was very difficult to find parameters for which the bivalent states became dominant. We found dominant bivalent states when beta, NURD and PR-DUB were all very small (of order 0.005; Fig. S5). Examination of categories and specific modifications showed that under these conditions, the system essentially becomes blocked in a state containing H3K27ac and H2Aub, as the rates of removal of these modifications are very low (Fig.S4b-g). Similar simulations in which beta was reduced to 0.005 and the rate of removal of H3K4 and H3K27 methylation by low KDM and UTX rates (of order 0.005) resulted in a dynamic bistable system that switches rapidly between silent (category 5) and silent bivalent (category 4) system states with a predominance of bivalent nucleosomes containing H3K4/K36 and H3K27 methylation (Fig. S5h-n). We did not find a parameter combination in which category 4 or category 2 stably dominated the system (Fig. S4, S5). We conclude that except under extreme conditions, the model avoids dominant bivalent states, preferring to pass via bistability in the transition between active and silent states.”

2.)

(a) Another simplification made in the model is to treat H3K4me3 and H3K36me3 as identical for the purpose of this model. Based on the observation that several mammalian H3K36 methyltransferases (or their SET domains rather than the full complexes) are inhibited by H2Aub, the authors state that trxG enzymes are inhibited by H2Aub. This is a potentially misleading generalization to make. To my knowledge, there is currently no evidence that H2Aub inhibits H3K4 methyltransferases. Given that H3K4 and H3K36 methyltransferases are very different in their architecture and regulation, the authors might want to reconsider this simplification. Other potentially very significant differences between H3K4 and H3K36 systems concern recruitment of the methyltransferases and removal of the marks. Generalizations about these aspects of the pathways are likely to be misleading as well.

Response 1:

We agree with the reviewer that the fusion of H3K4me and H3K36 me is vastly simplified with respect to the literature. However for several reasons we feel that this is a justifiable simplification for the purposes of modelling at the scale of detail we address. We have cited additional references and modified the text on Page 28 (Methods) to avoid giving the misleading impression that this is a feature of reality, rather than a simplifying feature of the model.

“Both H3K4me and H3K36me are associated with active genes, and fly and vertebrate ASH1 (ASH1L in vertebrates) and TRX (MLL1) show highly similar localisation on chromatin at many loci and interact directly with each other (in fly:³⁷, in vertebrates: ^{38,39}). TRX binding to chromatin is dependent on ASH1 at in flies ³⁶ and on H3K36me2 at several loci in vertebrates ³⁹. For simplicity in the model the methylated H3K4 and H3K36 were fused to a single species (H3K4/36), (see

Figure 1d, half-nucleosome states 1, 2, 5, 6, 9 and 10). Likewise the activities of ASH1 and TRX were fused in the model to a single activity (named TRXG in Figure 1d). We note that this simplification has costs, as situations in which the two proteins act independently of each other do exist^{29,58}. However the cost to the model of considering them separately would be large: The 3 dimensional model of Figure 1b would become four-dimensional, with a corresponding increase in unknown parameters. Furthermore any version of such a model that includes the interdependency of ASH1 and TRX would behave identically to one in which they are fused. Thus at present we consider that fusing the two is justified for the purposes of modelling but we do not wish to imply by this that they act in tandem in all possible situations.”

We also note that KDM2B demethylates both H3K4 and H3K36. (Table 1). We further discuss the limitations of this simplification. Page 21 (Discussing the comparison of asymmetry)

“We do not separate H3K4 and H3K36 methylation in the model. The fusion of these two modifications is justified by the biochemical data available, and has benefits in reducing the complexity of the model: the cube shown in Fig. 1b would be a four-dimensional structure if H3K4 and H3K36 were separated. However the cost of this reductionism is seen when comparing the model predictions to experimental data that show different behaviours of the two modifications. Further refinement of the model to treat H3K4 and H3K36 methylation separately will be interesting in future to address this discrepancy. “

(b) Moreover, the H2Aub-mediated inhibition has only been shown for mammalian enzymes, making the generalization towards *Drosophila* somewhat tenuous.

Response

Following the reviewer’s excellent suggestion to put literature in a table, (see Table 1) it becomes clear that there are several gaps in the literature for both fly and vertebrate. We make it clearer in the text, what has been shown for both, and what is missing. However we do not feel that the model is less applicable to flies than to vertebrates (there are the same number of gaps on both sides and these do not represent negative results but studies that have not been performed).

Page 9 (Results).

“Remarkably, we found that evidence exists for every enzymatic reaction (Table 1a), and for the majority of self-reinforcing and antagonistic interactions (Table 1b) in the model in both flies and vertebrates. Thus the model is potentially equally relevant to both.”

(c) The model would be equally appealing but based on stronger foundations if H3K36 was taken out of the equation. As bivalent domains occur at promoters, the authors should limit the model to modifications that occur at promoters and exclude H3K36 methylation.

Response

Given the above considerations, we wish to remain as comprehensive as possible at this stage. The domain we simulate is a generic domain of nucleosomes, in which all possible reactions could occur. The model can be modified to reflect any subset of interactions at any genomic location. Thus at this stage we have kept H3K36me in the model, and have given a clearer justification for this decision in the text (page 21 and 28, see above).

3.)

(a) It would be helpful to present the references that the different interactions, feedback loops, and inhibitory relationships in the model are based on in a table, giving references for both mammalian and *Drosophila* pathways in separate columns.

Response

We thank the reviewer for this excellent suggestion. We have summarised the literature as suggested in Table 1. We have kept the text descriptions in the Methods section, as these explain in detail how

each reaction as implemented in the model. We have clarified in the text where appropriate, which literature relates to fly and which to vertebrates.

(b) Demonstration of an interaction between Ash1L and MLL1 is being ascribed to Gregory et al. MCB 2007. However, this paper does not present any biochemical interaction data for these complexes and also states that Ash1L occupancy on chromatin is independent of MLL1.

Response

We thank the reviewer for pointing this out. We have cited more evidence for the interaction in vertebrates and modified text to clarify why we made this simplification. See response to 2(a) above.

(Ref 39: Zhu, L. *et al.*, ASH1L Links Histone H3 Lysine 36 Dimethylation to MLL Leukemia. *Cancer discovery* 6, 770-783 (2016).

(c) In general, care should be taken when discussing data on MLL methyltransferases. These have undergone several rounds of renaming, most notably MLL4 having been referred to as MLL2 previously. The authors should make sure that findings from older references are linked to the correct gene/protein. This should resolve some of the discrepancies discussed in Methods section 2.4. It is not clear why SET1 complexes in mammals were not mentioned.

Response

We thank the reviewer for pointing this out and have made sure that references to MLL2 and 4 are correct. However the dispute about the specificity of MLL1 for H3K4me1, 2, or 3 is not related to the MLL4/2 situation, as the relevant papers (Tie et al., 2014, Rickels et al., 2016), refer explicitly to MLL1 and TRX. This is a real discrepancy between two publications, and we cannot resolve it on the basis of available evidence. We have modified the text on page 28 (Methods) and in Table 1 to make this clearer. We also include a reference to SET1 in the text and in Table 1.

“Methylation of histone H3 on lysines 4 and 36 is associated with gene activation. The specificity of TRX and its vertebrate homolog MLL1 is disputed. Both have previously been reported to methylate H3K4me3⁹⁷⁻⁹⁹. However it was later shown that Most H3K4me3 is not catalysed by MLL1 in vertebrates^{100,101} and TRX in *Drosophila*^{102,103} but by the SET1/COMPASS complex. Two recent publications disagree on the specificity of TRX and MLL1. The first⁴² reports that both catalyse H3K4me1 only. The second¹⁰⁴ reports that both TRX and MLL1 are specific for H3K4me2. Some of these discrepancies may be attributable to anti H3K4me antibody specificity¹⁰⁵. However, for the purposes of our model, we are interested in any H3K4 methylation that genetically or biochemically antagonises PcG proteins, thus in the absence of a consensus and in the interests of simplification we refer to “H3K4me”.”

References cited:

- 42 Tie, F. *et al.*, Trithorax monomethylates histone H3K4 and interacts directly with CBP to promote H3K27 acetylation and antagonize Polycomb silencing. *Development* 141, 1129-1139 (2014).
- 97 Smith, S. T. *et al.*, Modulation of heat shock gene expression by the TAC1 chromatin-modifying complex. *Nat Cell Biol* 6, 162-167 (2004).
- 98 Milne, T. A. *et al.*, MLL targets SET domain methyltransferase activity to Hox gene promoters. *Molecular cell* 10, 1107-1117 (2002).
- 99 Nakamura, T. *et al.*, ALL-1 is a histone methyltransferase that assembles a supercomplex of proteins involved in transcriptional regulation. *Molecular cell* 10, 1119-1128 (2002).
- 100 Lee, J. H., Tate, C. M., You, J. S. & Skalnik, D. G., Identification and characterization of the human Set1B histone H3-Lys4 methyltransferase complex. *J Biol Chem* 282, 13419-13428 (2007).
- 101 Wu, M. *et al.*, Molecular regulation of H3K4 trimethylation by Wdr82, a component of human Set1/COMPASS. *Mol Cell Biol* 28, 7337-7344 (2008).
- 102 Ardehali, M. B. *et al.*, *Drosophila* Set1 is the major histone H3 lysine 4 trimethyltransferase with role in transcription. *The EMBO journal* 30, 2817-2828 (2011).

- 103 Hallson, G. *et al.*, dSet1 is the main H3K4 di- and tri-methyltransferase throughout Drosophila development. *Genetics* 190, 91-100 (2012).
- 104 Rickels, R. *et al.*, An Evolutionary Conserved Epigenetic Mark of Polycomb Response Elements Implemented by Trx/MLL/COMPASS. *Molecular cell* 63, 318-328 (2016).
- 105 Shah, R. N. *et al.*, Examining the Roles of H3K4 Methylation States with Systematically Characterized Antibodies. *Molecular cell* 72, 162-177 e167 (2018).

Reviewer #2 (Remarks to the Author):

Reviewer Comments for "Comprehensive theoretical analysis of the Polycomb – Trithorax system predicts that poised chromatin is robustly bistable and minimally bivalent".

This manuscript synthesizes a wealth of molecular and biological information about the PRC-TRX chromatin system into a model for predicting the distributions and dynamics of chromatin modifications at a small generic locus. The major claim of the paper is that chromatin states containing nucleosomes with both active and silent modifications (bivalent) are expected to be rare. Instead, this particular model predicts that chromatin states that switch back and forth between containing nucleosomes with fully active or fully silent modifications are to be expected. The model also predicts novel bivalent states that were not yet tested experimentally.

The subject of the manuscript is very timely. The chromatin/epigenetics field has been collecting a lot of data over the last few decades, and it is therefore ripe for complex models. This type of molecular simulations can nicely complement other efforts in the directions (such as machine learning on genome-wide data) in order to extracting general principles. Moreover, we appreciate the push the authors make for shifting the attention of the field from static snapshots of chromatin modifications to methods that could measure the dynamics of these states. This shift is especially needed for understanding poised genes, which by definition are genes where the dynamics of activation or silencing are supposed to be different.

The authors do an amazing job curating and summarizing the literature! That in itself is a big contribution. They also describe their model assumptions and parameters well, and actually make testable predictions.

While overall enthusiastic about the manuscript, we have some concerns about the assumptions that went into the model. Comparing the predictions of this model with other competing models in the field would greatly strengthen the manuscript. In addition, we have a few questions/suggestions about the definitions of terms in the paper and about choices made in the simulation parameters. Major comments:

1. The authors assume a very strong positive feedback between chromatin modifications: chromatin regulators bound at a single modified nucleosome can modify all the other nucleosomes in the array equally. This feedback contributes greatly to the dynamics bistability they predict. However, it is not clear how much experimental evidence there is to justify this strong feedback assumption. In fact, other models in the literature assume that the feedback only influences nucleosomes that are adjacent to the modified one (or that the strength of feedback decreases with distance):

-Berry 2017 (ref 11 in this manuscript),

-Hathaway et al., Dynamics and Memory of Heterochromatin in Living Cells, Cell (2012)

-Hodges et al., Dynamics of inherently bounded histone modification domains, PNAS (2012)

-Erdel et al, Generalized nucleation and looping model for epigenetic memory of histone modifications, PNAS (2016)

-see also Erdel, How Communication Between Nucleosomes Enables Spreading and Epigenetic Memory of Histone Modifications, BioEssays 2017 for a discussion on new experimental evidence for weak feedback .

Yes, it is true that some of the references above model HP1/H3K9 methylation. But the math is the same. And for either modification the feedback could be local, global (as assumed here), or in between.

It would be useful if the authors tried the other extreme assumption (feedback strictly between adjacent nucleosomes), and showed how their predictions change. This way, experimentalists can follow up and can actually discard one model or the other.

Response:

We thank the reviewer for pointing this out and drawing our attention to some really interesting work. We have modified the text in the discussion to clarify the experimental evidence supporting interactions over the short (2kb) distance that we model here, and have included a consideration of the work on longer distances in the discussion. However, we feel that a consideration of larger domains, although very interesting, would be outside the focus of the current work, and may be better placed for a separate study.

Discussion page 16:

“In particular, we treat each half – nucleosome in the array as equally likely to interact with its nearest neighbour as with any other in the array. We reason that since the model array is 2kb in size, this is a realistic approximation. PRC1-bound chromatin arrays have been shown to be highly compacted over similar distances⁵⁵ and ChIP-seq peaks of PcG and TrxG protein binding and the modifications they catalyse are typically at least 1-2kb in size^{25,56,57}. However some proteins and modifications do spread over much longer distances of several tens of kilobases^{25,58}. Several theoretical and experimental studies have examined the effects of distance between nucleosomes in large modelled arrays, and the phenomena of spreading and looping^{8,53,54,59,60}. It will be interesting in future to extend the model to address larger domains. “

Citations: Experimental

- 25 Mikkelsen, T. S. *et al.*, Genome-wide maps of chromatin state in pluripotent and lineage-committed cells. *Nature* 448, 553-560 (2007).
- 55 Francis, N. J., Kingston, R. E. & Woodcock, C. L., Chromatin compaction by a polycomb group protein complex. *Science* 306, 1574-1577 (2004).
- 56 Peng, J. C. *et al.*, Jarid2/Jumonji coordinates control of PRC2 enzymatic activity and target gene occupancy in pluripotent cells. *Cell* 139, 1290-1302 (2009).
- 57 Schuettengruber, B. *et al.*, Functional anatomy of polycomb and trithorax chromatin landscapes in *Drosophila* embryos. *PLoS biology* 7, e13 (2009).
- 58 Schwartz, Y. B. *et al.*, Alternative epigenetic chromatin states of polycomb target genes. *PLoS Genet* 6, e1000805 (2010).

Citations: Theoretical

- 8 Dodd, I. B., Micheelsen, M. A., Sneppen, K. & Thon, G., Theoretical analysis of epigenetic cell memory by nucleosome modification. *Cell* 129, 813-822 (2007).
- 53 Erdel, F. & Greene, E. C., Generalized nucleation and looping model for epigenetic memory of histone modifications. *Proceedings of the National Academy of Sciences of the United States of America* 113, E4180-4189 (2016).
- 54 Obersriebnig, M. J., Pallesen, E. M., Sneppen, K., Trusina, A. & Thon, G., Nucleation and spreading of a heterochromatic domain in fission yeast. *Nature communications* 7, 11518 (2016).
- 59 Hathaway, N. A. *et al.*, Dynamics and memory of heterochromatin in living cells. *Cell* 149, 1447-1460 (2012).
- 60 Erdel, F., How Communication Between Nucleosomes Enables Spreading and Epigenetic Memory of Histone Modifications. *BioEssays : news and reviews in molecular, cellular and developmental biology* 39 (2017).

2. The definition the authors choose for bivalent nucleosomes is very restrictive, and might artificially make it seem like they are very rare. The authors define bivalent nucleosomes using half nucleosomes: the opposing modifications have to be on the same half of the nucleosome to call it bivalent. This is not the standard definition, it is not what is usually measured experimentally (see Shema, 2016, ref 33 and Weiner 2016, ref 28), and it might not be what matters for effectors biologically. More commonly, bivalent nucleosomes could have the opposing modifications on the same half or on opposing halves of the same nucleosome. How common are these type of bivalent nucleosomes with modifications on the same nucleosome (on either side) in the results of the simulations? If they are quite common, it could change the conclusions of the paper.

Response

We thank the reviewer for this excellent suggestion and we agree it makes total sense. We have now adapted the model to consider whole nucleosomes (Figure 2, Figure S1). In all subsequent figures, the system state is evaluated in terms of whole nucleosomes. The conclusions are very similar. Remarkably, despite the ability to adopt many more bivalent states, we find that the key conclusions of the original manuscript still hold true when considering whole nucleosomes. **The system remains robustly bistable and minimally bivalent.**

(See also response to reviewer 1, point 1.)

3. I believe it would improve the paper to make the definition of poised clearer and more explicit from the very beginning (line 8 on page 4 for example). Especially given that unfortunately, in the literature often poised and bivalent are used interchangeably. Is poised supposed to imply the ability to rapidly switch from a low/off expression state to an on state upon stimulation (change in parameters)? The authors did not explore this.

Response:

We thank the reviewer for drawing our attention to this. We have clarified in several places, including the introduction page 4 line 8, that we use “poised” to mean “undecided”.

Page 4 line 8: “Bivalent chromatin is thought to represent a poised or “undecided” form..”

Page 5 line 22: “Importantly, midway in the transition between active and silent states, “poised” chromatin is not bivalent in the model, but is robustly bistable, and differs from monostable modes only in its higher frequency of switching.”

4. The authors only simulate 10 nucleosomes. Is that enough? Domains of modified nucleosomes are at least an order of magnitude larger than that (15kb or larger, so 100 nucleosomes or larger). So are gene bodies in mammalian cells. How would the conclusions change if the authors simulated a larger nucleosomal array?

Response:

See response to point 1 above. As shown in Dodd et al., 2007, larger domains would essentially lead to a more stable system, requiring longer simulations for a switch. As discussed above we feel this is a very interesting area, but beyond the scope of the current study.

5. It would be interesting to see the single nucleosome trajectories in the simulations. Are bivalent nucleosomes common intermediates in the bistable switching?

Response:

For a single nucleosome, a variety of paths can enable it to switch from an active to a silent state (see Fig 2). The most favoured transition for half - nucleosomes is via the unmodified state. Given the complexity of the new whole nucleosome model, we have not included single nucleosome trajectories in the revised manuscript, in order to keep the discussion at the level of global nucleosome array properties. We have tried to make clearer that the active and silent states are a dynamic mix of “pure” active or silent, and bivalent active or silent (see new Figure 3, 4 and 5c). We did not find an increase of bivalent states in the bistable transition zone except under extreme conditions (Figure S5).

6. It would improve the paper (and reproducibility) if some of the simulation details are better justified or documented. For instance:

a. how was the number of steps in the simulation chosen? Based on convergence?

Response:

Yes, this was based on convergence.

b. In Supplementary Table 1, please label what the parameters in the columns represent (in a way that can easily be connected with the methods section, so that results can be reproduced). Presumably these are rates.

Response:

Yes, they are rates, this has been added to the Table S1.

7. How do the results of single-molecule Co-ChIP (Weiner 2016, ref 28), which analyzed a lot of combinations of modifications, compare with the predictions of the model?

Response:

Weiner et al (now ref. 29) did not look at H2Aub, but they did evaluate co- occurrence of H3K4me, H3K36me, and H3K27me. They found strong co occurrence of H3K4me1, 2 and 3 with H3K27me on the most silent class of genes, in agreement with our prediction of “silent bivalent” (this is cited in the discussion on page 23, line 16).

“Several vertebrate studies have noted a correlation between K4/K27me bivalent chromatin and silent gene expression states in ESC cells and also in differentiated cell types, consistent with the model prediction ^{21-25,29,32-34,51}.”

Weiner et al also found that co – occurrence of H3K27me3 and H3K27ac is virtually undetectable, consistent with our model prediction (even with whole nucleosomes). We point this out on p22 line 11.

“In contrast, the co – occurrence of H3K27ac and H3K27me3 on the same nucleosome, even on opposite tails, is predicted to be rare (Fig. 4c), consistent with experimental analysis by co-ChIP ²⁹.”

In addition we compare our results to two other recent publications that quantitatively evaluated bivalent heterogeneity. We present results in Figure S7 and text on page 23 (Results section). Ref. 30: Kinkley et al., 2016; Ref. 75: Grzybowski 2015.

“...two recent studies have addressed heterogeneity within bivalent chromatin carrying H3K27me3 and H3K4me3^{30,75}. Both studies identified bivalent chromatin carrying higher amounts of H3K4me3 than H3K27me3 associated with active loci, and the converse (higher H3K27me3 than H3K4me3) associated with silent loci. Consistent with these observations, our model does indeed predict that the ratio of H3K27me to H3K4me in the population of bivalent nucleosomes carrying both marks is reduced as the system switches from the silent to the active regime (Fig. S7e). It would be highly interesting in future to apply the approaches of ^{30,75} to the H3K27ac/H2Aub form, which we predict to display the opposite behaviour (i.e. the proportion of H3K27ac to H2Aub is expected to decrease as the system becomes silent (Fig. S7d).)”

Minor comments:

1. References to Figure 2D and 2E on page 13, lines 1 and 2 should be changed to Figure 3D and 3E, respectively.

2. On the following pages/lines references to Figure 1C should be changed to Figure 1D: pg 21 line 24; pg 22 lines 5, 14; pg 23 lines 5, 11, 22; pg 24 line 8; pg 25 lines 15, 17, 19, 21; pg 27 lines 7, 15, 23; pg 29 line 10.

Response:

All figure references have now been rechecked and adapted to new figure numbers.

3. The blue circles in Figure S2 can be difficult to distinguish from the black circles; perhaps changing the color, for example to cyan, would make this figure easier to interpret.

Response:

All figures with blue dots have been changed to cyan.

REVIEWERS' COMMENTS:

Reviewer #1 (Remarks to the Author):

The revised version of 'Comprehensive theoretical analysis of the Polycomb – Trithorax system predicts that poised chromatin is robustly bistable' constructively addresses the majority of the reviewers' comments, resulting in a much-improved manuscript that should be fit for publication in Nature Communications.

The fusion of H3K4me and H3K36me remains a significant concern, but I accept the arguments presented by the authors explaining their decision to fuse these two marks. Moreover, the authors are clear about their rationale and the potential consequences of the fusion in the manuscript, allowing the reader to take this into account. As a minor point, in the newly added parts of the discussion related to this issue (page 21), the authors state that H3K36me₂ was detected on 25% of H3K27me_{2/3} tails (ref 32). This does not appear to be correct – from the figures in ref 32 it appears to be a few percent at best. Moreover, Schmitges et al (ref 44) also reported inhibition of PRC2 by both H3K36me₃ and H3K36me₂, and Young et al. (Mol Cell Proteomics 2009) reported that the fraction of H3 carrying both H3K27me₃ and H3K36me₃ as 0.078% and that of H3K27me₃ and H3K36me₂ 1.315% of H3 molecules. The authors should correct this statement accordingly.

In Figure 5, the authors predict outcomes for ChIP, reChIP, and mRNA expression analysis for bivalent versus bistable systems. It would be helpful if the authors could expand a bit more on the rationale behind some of their predictions, most crucially why a H3K27/H3K4 bivalent state (Fig 5g) is predicted to result in intermediate mRNA expression on a single cell level rather than low or no mRNA expression, as is indeed observed for most bivalent genes on a population level. What is the molecular basis for this prediction? Also, why would such a state not be able to generate heterogeneity in expression across a population? Switching from a bivalent to a fully active or fully silent state of a nucleosome would be just as feasible as directly switching between a fully active or fully silent nucleosome state. Moreover, it is not clear on what molecular basis a H2Aub/H3K27ac state is predicted to result in high single cell mRNA levels. I think it would be important for the authors to be as clear as possible about these predictions, as they are key, testable hypotheses that undoubtedly many readers of the manuscript will take into account in designing experiments and in assessing the validity of the model presented here.

Reviewer #2 (Remarks to the Author):

The authors addressed the main concern we had with the paper by going from half nucleosomes to full nucleosomes. They significantly changed the main figures, and re-wrote a significant portion of the paper. Moreover, the authors included Table 1, which is extremely useful for evaluating the assumptions of the model. Together, this effort made the manuscript much stronger!

In addition, the authors better connected their predictions with existing experimental evidence for frequencies of different types of bivalent modifications, and expectations from single-cell measurements.

We agree the larger size array is beyond the scope of this work, and we find appropriate the manner in which this issue is addressed as a possible caveat/future direction in the discussion section. Indeed, a larger array would require not just an increase in simulation size, but a modulation of the probability that one nucleosome affects other nucleosomes in the array. Maybe this point could be more explicitly spelled out in that discussion section. Right now this coupling parameter is set to 1, but it would have to decrease as a function of genomic distance. The authors argue that within 10 nucleosomes (1-2kb), having this parameter set to 1 is justified, and we didn't find experimental evidence to suggest otherwise.

Reviewers' comments:**Reviewer #1 (Remarks to the Author):**

The revised version of 'Comprehensive theoretical analysis of the Polycomb – Trithorax system predicts that poised chromatin is robustly bistable' constructively addresses the majority of the reviewers' comments, resulting in a much-improved manuscript that should be fit for publication in Nature Communications.

The fusion of H3K4me and H3K36me remains a significant concern, but I accept the arguments presented by the authors explaining their decision to fuse these two marks. Moreover, the authors are clear about their rationale and the potential consequences of the fusion in the manuscript, allowing the reader to take this into account. As a minor point, in the newly added parts of the discussion related to this issue (page 21), the authors state that H3K36me₂ was detected on 25% of H3K27me_{2/3} tails (ref 32). This does not appear to be correct – from the figures in ref 32 it appears to be a few percent at best.

Response: We thank the reviewer for pointing this out. We realise that we mistakenly referred to the co-occurrence of H3K27me_{2/3} and H3K36me₂ on whole nucleosomes in reference 32 (shown e.g. in Figure 4D) and not on the same tail. We were unable to find a quantification of the co- occurrence of these marks on the same tail in reference 32, and have removed the citation from the discussion on page 21, line 17.

Moreover, Schmitges et al (ref 44) also reported inhibition of PRC2 by both H3K36me₃ and H3K36me₂,

Response: We cited Schmitges et al for this observation on page 32, line 21.

and Young et al. (Mol Cell Proteomics 2009) reported that the fraction of H3 carrying both H3K27me₃ and H3K36me₃ as 0.078% and that of H3K27me₃ and H3K36me₂ 1.315% of H3 molecules. The authors should correct this statement accordingly.

Response: We thank the reviewer for drawing our attention to the paper of Young et al., which we were not aware of. We have cited this work in the discussion on page 21, lines 15 to 18 (reference 52). The reviewer correctly points out that the total fractions of H3 nucleosomes carrying combinations of marks are low. However the fraction we are interested in is the percentage of H3K27me₃ modified tails that also carry H3K36me₂. We have analysed the data in Table 1 of Young et al., by summing up all occurrences of H3K27me₃ in which H3K36 is not methylated or is monomethylated, and comparing this to the occurrences of H3K27me₃ together with H3K36me₂ or me₃. This enables a calculation of the % of H3K27me₃ tails that additionally carry H3K36me₂, which is the number we are interested in. This is calculated at 3.9% and is mentioned in the revised manuscript on page 21.

In Figure 5, the authors predict outcomes for ChIP, reChIP, and mRNA expression analysis for bivalent versus bistable systems. It would be helpful if the authors could expand a bit more on the rationale behind some of their predictions, most crucially why a H3K27/H3K4 bivalent state (Fig 5g) is predicted to result in intermediate mRNA expression on a single cell level rather than low or no mRNA expression, as is indeed observed for most bivalent genes on a population level. What is the molecular basis for this prediction? Also, why would such a state not be able to generate heterogeneity in expression across a population? Switching

from a bivalent to a fully active or fully silent state of a nucleosome would be just as feasible as directly switching between a fully active or fully silent nucleosome state.

Response: We thank the reviewer for pointing out that Figure 5 was unclear. We previously referred to the “bivalent model” as the current idea from literature, and the “bistable model” as the theoretical one that emerges from the current work. We realize that this was confusing and we have simplified Figure 5 to refer only to the predictions of the bistable model. The reviewer’s comments on the potential behavior of bivalent chromatin are valid questions, but none of these behaviours emerge from our model. We hope that by restricting Figure 5 to the predictions of the new mathematical model we have clarified this point.

Moreover, it is not clear on what molecular basis a H2Aub/H3K27ac state is predicted to result in high single cell mRNA levels. I think it would be important for the authors to be as clear as possible about these predictions, as they are key, testable hypotheses that undoubtedly many readers of the manuscript will take into account in designing experiments and in assessing the validity of the model presented here.

Response: We have shown that the presence of H2Aub/H3K27ac bivalent nucleosomes is strongly correlated with active system states (Figure 4, Supplementray Figure 6). As we point out in the discussion on page 20, line 3-4: “Although we are cautious not to equate chromatin state directly with transcriptional state, we do expect some correlation.”

Our reasoning for predicting that the presence of H2Aub/H3K27ac will correlate with high single cell mRNA levels is thus as follows:

- 1) H2Aub/H3K27ac bivalent nucleosomes are correlated with active system states.
- 2) Active system states are expected to correlate with high transcription levels.
- 3) Therefore H2Aub/H3K27ac bivalent nucleosomes should correlate with high transcription levels.

We make no claims as to molecular cause and consequence.

Reviewer #2 (Remarks to the Author):

The authors addressed the main concern we had with the paper by going from half nucleosomes to full nucleosomes. They significantly changed the main figures, and re-wrote a significant portion of the paper. Moreover, the authors included Table 1, which is extremely useful for evaluating the assumptions of the model. Together, this effort made the manuscript much stronger!

In addition, the authors better connected their predictions with existing experimental evidence for frequencies of different types of bivalent modifications, and expectations from single-cell measurements.

We agree the larger size array is beyond the scope of this work, and we find appropriate the manner in which this issue is addressed as a possible caveat/future direction in the discussion section. Indeed, a larger array would require not just an increase in simulation size, but a modulation of the probability that one nucleosome affects other nucleosomes in the array. Maybe this point could be more explicitly spelled out in that discussion section. Right now this coupling parameter is set to 1, but it would have to decrease as a function of genomic distance. The authors argue that within 10 nucleosomes (1-2kb), having this

parameter set to 1 is justified, and we didn't find experimental evidence to suggest otherwise.

Response: We thank the reviewer for constructive feedback, and have added the sentence in the discussion on page 17 line 8-9:

“for example by modulating the probability that one nucleosome affects others in the array.”

Reviewer #2 (Remarks to the Author):

—

—